# A transient role of the ciliary gene *Inpp5e* in controlling direct versus indirect neurogenesis in cortical development

Kerstin Hasenpusch-Theil[1,2], Christine Laclef[3†], Matt Colligan[1], Eamon Fitzgerald[1], Katherine Howe[1], Emily Carroll[1], Shaun R Abrams[4], Jeremy F Reiter[4,5], Sylvie Schneider-Maunoury[3], Thomas Theil[1,2]*

[1]Centre for Discovery Brain Sciences, University of Edinburgh, Edinburgh, United Kingdom; [2]Simons Initiative for the Developing Brain, University of Edinburgh, Edinburgh, United Kingdom; [3]Sorbonne Université, CNRS UMR7622, INSERM U1156, Institut de Biologie Paris Seine (IBPS) - Developmental Biology Unit, Paris, France; [4]Department of Biochemistry and Biophysics, University of California, San Francisco, San Francisco, United States; [5]Chan Zuckerberg Biohub, San Francisco, United States

*For correspondence:
thomas.theil@ed.ac.uk

Present address: †Sorbonne Université, Institut du Fer à Moulin INSERM U1270, Paris, France

Competing interests: The authors declare that no competing interests exist.

**Abstract** During the development of the cerebral cortex, neurons are generated directly from radial glial cells or indirectly via basal progenitors. The balance between these division modes determines the number and types of neurons formed in the cortex thereby affecting cortical functioning. Here, we investigate the role of primary cilia in controlling the decision between forming neurons directly or indirectly. We show that a mutation in the ciliary gene *Inpp5e* leads to a transient increase in direct neurogenesis and subsequently to an overproduction of layer V neurons in newborn mice. Loss of *Inpp5e* also affects ciliary structure coinciding with reduced Gli3 repressor levels. Genetically restoring Gli3 repressor rescues the decreased indirect neurogenesis in *Inpp5e* mutants. Overall, our analyses reveal how primary cilia determine neuronal subtype composition of the cortex by controlling direct versus indirect neurogenesis. These findings have implications for understanding cortical malformations in ciliopathies with *INPP5E* mutations.

## Introduction

Building a functional cerebral cortex which confers humans with their unique cognitive capabilities requires controlling the proliferation of neural progenitor cells and the timing and modes of neurogenic cell divisions. Varying the timing and modes of neurogenesis affects neuronal numbers and subtype composition of the cortex (*Florio and Huttner, 2014*). In the developing murine cortex, radial glial cells (RGCs) represent the major neural stem cell type. Residing in the ventricular zone, they express Pax6 and undergo interkinetic nuclear migration dividing at the ventricular surface (*Götz et al., 1998*; *Warren et al., 1999*). Initially, RGCs go through rounds of symmetric proliferative divisions to produce two RGCs increasing the progenitor pool but switch to asymmetric divisions at the beginning of cortical neurogenesis. RGCs generate neurons in two ways, either directly or indirectly via the production of basal progenitors (BPs) that settle in the subventricular zone (SVZ) and express the Tbr2 transcription factor (*Englund et al., 2005*). In the mouse, the majority of BPs divide once to produce two neurons, whereas the remainders undergo one additional round of symmetric proliferative division before differentiating into two neurons (*Haubensak et al., 2004*; *Miyata, 2004*; *Noctor et al., 2004*). In this way, BPs increase neuron output per RGC and have therefore been implicated in the evolutionary expansion of the mammalian cerebral cortex

(*Martínez-Cerdeño et al., 2006*). Thus, the balance between direct and indirect neurogenesis is an important factor in generating appropriate neuron numbers and types.

The mechanisms that fine tune this balance and thereby adjust the numbers and types of neurons produced in the cortex have only recently been investigated. Mitotic spindle orientation (*Postiglione et al., 2011*) and endoplasmic reticulum (ER) stress (*Gladwyn-Ng et al., 2018*; *Laguesse et al., 2015*) are contributing factors to control the generation of basal progenitors. In addition, levels of Slit/Robo and Notch/Delta signaling were shown to be evolutionarily conserved factors that determine the predominant mode of neurogenesis (*Cárdenas et al., 2018*). Moreover, feedback signals from postmitotic neurons control the fate of radial glial daughter cells involving the release of Neurotrophin-3 and Fgf9 (*Parthasarathy et al., 2014*; *Seuntjens et al., 2009*) as well as the activation of a Notch-dependent signaling pathway (*Wang et al., 2016*). These studies highlight the importance of cell-cell signaling in controlling the cell lineage of cortical progenitors (*Silva et al., 2019*) and emphasize the necessity of studying the cellular mechanisms by which these signals control the decision by RGCs to undergo direct or indirect neurogenesis.

Given the importance of cell-cell signaling, it is likely that the primary cilium, a signaling hub in embryogenesis in general and in neural development in particular (*Valente et al., 2014*), plays key roles in determining the balance between direct versus indirect neurogenesis. The cilium is a subcellular protrusion that predominately emanates from the apical surface of RGCs projecting into the ventricular lumen. The phenotypes of several mouse lines mutant for ciliary genes underline the importance of the primary cilium in forebrain development but these mutants often suffer from severe patterning defects (*Ashique et al., 2009*; *Besse et al., 2011*; *Willaredt et al., 2008*) which make elucidating ciliary roles in determining the lineage of cortical progenitors difficult. To address how cilia control cortical progenitor development, we investigated corticogenesis in a mouse mutant for the ciliary gene *Inpp5e*.

*INPP5E* is mutated in Joubert syndrome (JS) (*Bielas et al., 2009*; *Jacoby et al., 2009*), a ciliopathy characterized by cerebellar defects in which a subset of patients also shows malformations of the cerebral cortex including heterotopias, polymicrogyria and agenesis of the corpus callosum (*Valente et al., 2014*). *Inpp5e* encodes Inositol polyphosphate 5 phosphatase E, an enzyme that is localized in the ciliary membrane and that hydrolyses the phosphatidylinositol polyphosphates PI(4,5)$P_2$ and PI(3,4,5)$P_3$ (*Bielas et al., 2009*; *Jacoby et al., 2009*). In this way, it controls the inositol phosphate composition of the ciliary membrane and thereby regulates the activity of several signaling pathways and cilia stability (*Bielas et al., 2009*; *Chávez et al., 2015*; *Garcia-Gonzalo et al., 2015*; *Jacoby et al., 2009*; *Plotnikova et al., 2015*). In contrast to *Inpp5e*'s extensively studied biochemical and cellular roles, little is known how these diverse functions are employed at the tissue level to control RGC lineage.

Here, we show that loss of *Inpp5e* function results in an increase in neuron formation at the expense of basal progenitor production in the E12.5 cortex and in an overproduction of Ctip2+ layer V neurons in newborn mutants. Moreover, RGC cilia show unusual membranous structures and/or abnormal numbers of microtubule doublets affecting the signaling capabilities of the cilium. The levels of Gli3 repressor (Gli3R), a critical regulator of cortical stem cell development (*Hasenpusch-Theil et al., 2018*; *Wang et al., 2011*), is reduced and re-introducing Gli3R rescues the decreased formation of basal progenitors. Taken together, these findings implicate *Inpp5e* and the primary cilium in controlling the decision of RGCs to either undergo direct neurogenesis or to form basal progenitors, thereby governing the neuronal subtype composition of the cerebral cortex.

## Results

### *Inpp5e*$^{\Delta/\Delta}$ embryos show mild telencephalic patterning defects

Controlling the balance between direct and indirect neurogenesis in the developing cerebral cortex is mediated by cell-cell signaling (*Cárdenas et al., 2018*) and hence may involve the primary cilium. To investigate potential ciliary roles, we started characterizing cortical stem cell development in embryos mutant for the *Inpp5e* gene which has a prominent role in ciliary signaling and stability. Mutations in ciliary genes have previously been found to result in telencephalon patterning defects, most notably in a ventralization of the dorsal telencephalon and/or in defects at the corticoseptal (CSB) and pallial/subpallial boundaries (PSPB) (*Ashique et al., 2009*; *Besse et al., 2011*;

*Willaredt et al., 2008*). Therefore, we first considered the possibility that such early patterning defects may be present in *Inpp5e* mutant embryos and could affect cortical stem cell development. In situ hybridization and immunofluorescence analyses of E12.5 control and *Inpp5e*$^{\Delta/\Delta}$ embryos revealed no obvious effect on the expression of dorsal and ventral telencephalic markers at the corticoseptal boundary (*Figure 1—figure supplement 1A–F*). In contrast, the pallial/subpallial boundary was not well defined with a few scattered Pax6+ and *Dlx2* expressing cells on the wrong side of the boundary, that is in the subpallium and pallium, respectively (*Figure 1—figure supplement 1G–L*). Moreover, the hippocampal anlage appeared smaller and disorganized with low level and diffuse expression of cortical hem markers (*Figure 1—figure supplement 2*), consistent with known roles of Wnt/β-catenin and Bmp signaling in hippocampal development (*Galceran et al., 2000*; *Lee et al., 2000*). In contrast, progenitors in the neocortical ventricular zone of *Inpp5e* mutant mice expressed the progenitor markers *Emx1*, *Lhx2*, *Pax6* and *Ngn2,* though the levels of Pax6 protein expression appeared reduced in the medial neocortex suggestive of a steeper lateral to medial Pax6 expression gradient in mutant embryos. These expression patterns were maintained in E14.5 *Inpp5e*$^{\Delta/\Delta}$ embryos but revealed an area in the very caudal/dorsal telencephalon where the neocortex was folded (*Figure 1—figure supplement 3*). These folds became more prominent at more caudal levels and were also present in the hippocampal anlage. Taken together, these findings indicate that *Inpp5e* mutants have mild patterning defects affecting the integrity of the PSPB, hippocampal development and the caudal-most neocortex while the rostral neocortex shows no gross malformation or mispatterning and can therefore be analyzed for effects of the *Inpp5e* mutation on direct and indirect neurogenesis.

### *Inpp5e* controls direct vs indirect neurogenesis in the lateral neocortex

Based upon these findings, we started analyzing the proliferation and differentiation of RGCs in *Inpp5e*$^{\Delta/\Delta}$ embryos in the rostrolateral and rostromedial neocortex to avoid the regionalization defects described above. As a first step, we determined the proportion of RGCs, basal progenitors and neurons in these regions in E12.5 embryos. Double immunofluorescence for PCNA which labels all progenitor cells (*Hall et al., 1990*) and the radial glial marker Pax6 did not reveal differences in the proportions of RGCs at both medial and lateral levels (*Figure 1A–D*). In contrast, the proportion of Tbr2+ basal progenitors was reduced laterally but not medially (*Figure 1E–H*). This decrease coincided with an increase in Tbr1+ neurons specifically in the lateral neocortex (*Figure 1I–L*).

To determine whether these alterations are maintained at a later developmental stage, we repeated this investigation in E14.5 embryos. This analysis revealed no significant differences in the proportion of Pax6+ RGCs (*Figure 2A–D*). Similarly, there was no alteration in the proportion of Tbr2+ basal progenitors in lateral neocortex; however, their proportion was reduced medially (*Figure 2E–H*). To label cortical projection neurons, we used double immunofluorescence for Tbr1 and Ctip2 which allowed us to distinguish between Tbr1+Ctip2+ and Tbr1-Ctip2+ neurons. Quantifying these subpopulations showed no effect on the formation of Tbr1+ Ctip2+ neurons in *Inpp5e*$^{\Delta/\Delta}$ embryos. In contrast, the proportion of Tbr1- Ctip2+ neurons was reduced medially but increased in the lateral neocortex (*Figure 2I–N*). Taken together with our E12.5 analyses, these findings show that in the lateral neocortex of *Inpp5e*$^{\Delta/\Delta}$ an increase in the proportion of Tbr1+ neurons at E12.5 is followed by an augmented fraction of Tbr1- Ctip2+ neurons at E14.5, whereas the proportion of basal progenitors recovered after an initial down-regulation.

To address the defective cellular processes underlying these neurogenesis defects in *Inpp5e* mutants, we first investigated programmed cell death and found few apoptotic cells in the control and mutant cortex (*Figure 3—figure supplement 1*). Next, we measured proliferation rates of cortical progenitors and performed double immunofluorescence for PCNA and pHH3 which labels mitotic RGCs located at the ventricular surface and dividing basal progenitors in abventricular positions (*Figure 3—figure supplement 2*). This analysis revealed no statistically significant differences in the E12.5 and E14.5 lateral neocortex of control and *Inpp5e*$^{\Delta/\Delta}$ embryos. The proportion of mitotic apical and basal progenitors, however, was reduced in the E12.5 medial neocortex (*Figure 3—figure supplement 2*). Interestingly, this decrease in the fraction of mitotic basal progenitors precedes the reduced proportion of basal progenitors in the E14.5 medial neocortex (*Figure 2E–H*).

The cell cycle represents another key regulator of neuronal differentiation and a mutation in *Kif3a* affects ciliogenesis and the cell cycle in the developing neocortex (*Wilson et al., 2012*). To investigate the possibility of altered cell cycle kinetics, we used a BrdU/IdU double labeling strategy

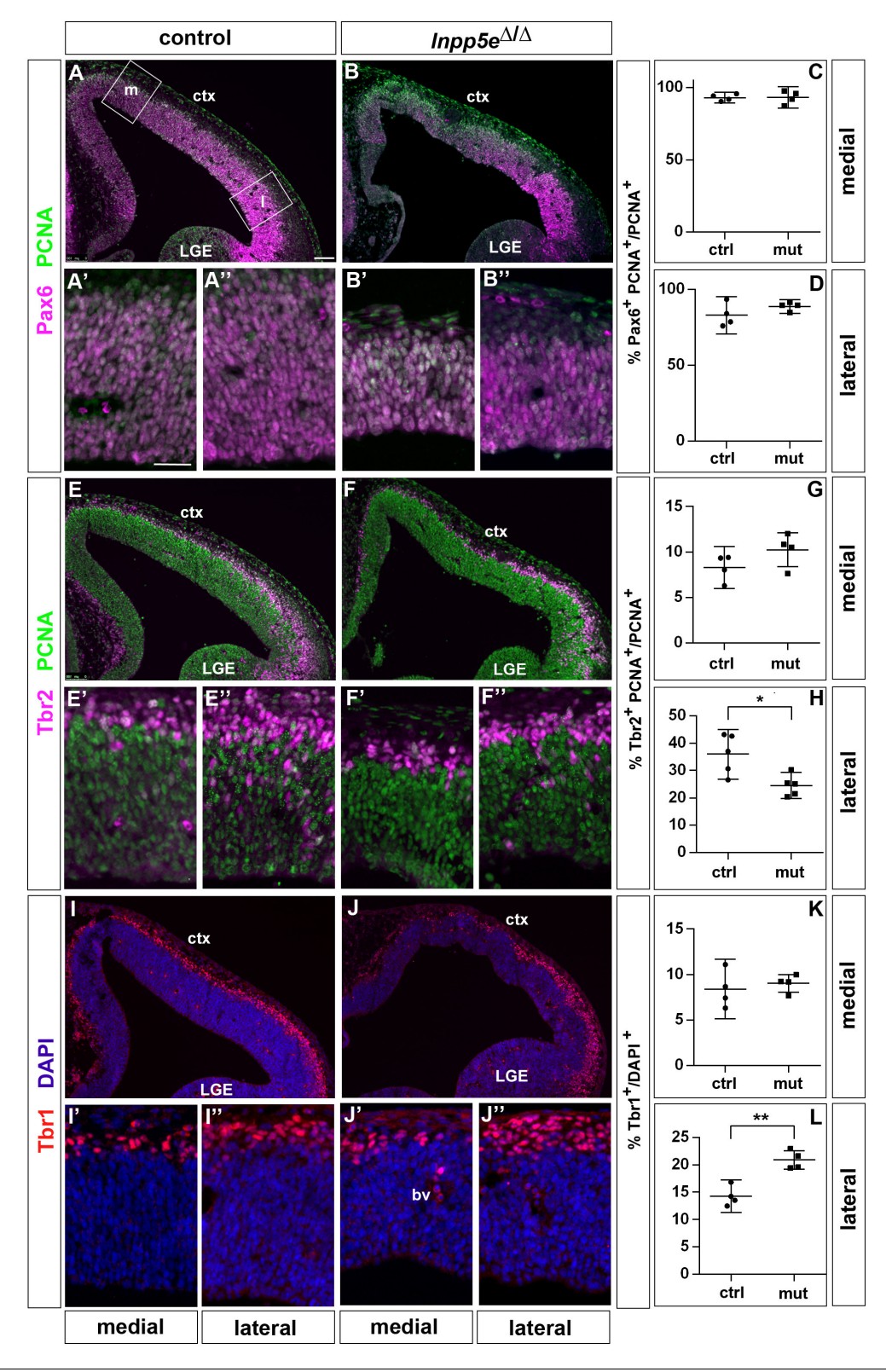

**Figure 1.** Increased neuron formation in the dorsolateral telencephalon of E12.5 *Inpp5e*^ΔlΔ embryos. (**A–D**) Pax6/PCNA double immunofluorescence staining revealed the proportion of apical radial glial cells which remained unaltered in the mutant. The boxes in (**A**) indicate the regions in the medial (m) and lateral (l) telencephalon at which cell counts were performed. (**E–H**) Reduced proportions of basal progenitors in the lateral telencephalon as revealed by staining for Tbr2 and PCNA. (**I–L**) Tbr1 immunostaining showed that the proportion of neurons is increased in the lateral telencephalon. (**A–**

*Figure 1 continued on next page*

*Figure 1 continued*

J) The insets labelled with ' and " are representative magnifications of medial and lateral levels, respectively. All statistical data are presented as means ± 95% confidence intervals (CI); unpaired t-tests; n = 4 except for (H) with n = 5; *p<0.05; **p<0.01. Scale bar: 100 µm (A) and 50 µm (A'). ctx: cortex; LGE: lateral ganglionic eminence.

The online version of this article includes the following figure supplement(s) for figure 1:

**Figure supplement 1.** Formation of the telencephalic boundaries in *Inpp5e*$^{\Delta/\Delta}$ embryos.

**Figure supplement 2.** Wnt/β-catenin and Bmp signaling in the dorsomedial telencephalon of E12.5 *Inpp5e*$^{\Delta/\Delta}$ embryos.

**Figure supplement 3.** Expression of cortical progenitor markers in *Inpp5e*$^{\Delta/\Delta}$ embryos.

(*Martynoga et al., 2005*; *Nowakowski et al., 1989*) to determine S phase length and total cell cycle length in RGCs and found no statistically significant changes in these parameters (*Figure 3—figure supplement 3*).

Finally, the increased neuron production could also be explained by an increase in direct neurogenesis at the expense of basal progenitor cell formation. To test this possibility, we gave BrdU to E11.5 pregnant mice 24 hr before dissecting the embryos. We then used BrdU immunostaining in conjunction with Tbr1 and Tbr2 to identify the neurons and basal progenitors formed in the lateral neocortex within the 24 hr time period. This analysis showed that the proportion of Tbr1+ neurons compared to the total number of BrdU+ cells increased while the Tbr2+ proportion decreased in *Inpp5e* mutants (*Figure 3*). Since the cell cycle of basal progenitors is longer than 24 hr (*Arai et al., 2011*), the 24 hr interval used in our cell cycle exit experiment was too short for newly formed basal progenitors to undergo one additional round of the cell cycle and as the BrdU label would have been diluted with a further round of division, this analysis supports our hypothesis that direct neurogenesis became more prevalent in *Inpp5e*$^{\Delta/\Delta}$RGCs.

## Cortical malformations in *Inpp5e*$^{\Delta/\Delta}$ embryos

Next, we investigated the consequences of this increase in direct neurogenesis on cortical size and layer formation. Since *Inpp5e*$^{\Delta/\Delta}$ newborn pups die perinatally (*Bielas et al., 2009*), we focused our analysis on E18.5 embryos. The mutant lacked obvious olfactory bulbs, as revealed by whole mounts of control and mutant brains (*Figure 4—figure supplement 1*). To gain insights into the overall histology of the mutant forebrain, we stained coronal sections with DAPI. This analysis showed that most of the mutant cortex was thinner except for the rostrolateral level (*Figure 4—figure supplement 2*). In addition, the hippocampus was malformed with a smaller dentate gyrus. Investigating the expression of markers characteristic of the entire hippocampus (*Nrp2*; *Galceran et al., 2000*), the CA1 field (Scip1; *Frantz et al., 1994*) and the dentate gyrus (Prox1; *Oliver et al., 1993*) showed that these hippocampal structures were present but were severely reduced in size and disorganized in *Inpp5e*$^{\Delta/\Delta}$ embryos (*Figure 4—figure supplement 3*). In addition, the corpus callosum, the major axon tract connecting the two cerebral hemispheres, was smaller. We confirmed this effect by staining callosal axons and surrounding glial cells that guide these axons to the contralateral hemisphere with L1 and GFAP, respectively (*Figure 4—figure supplement 4*).

After characterizing the gross morphology of the *Inpp5e*$^{\Delta/\Delta}$ cortex, we next investigated whether the increased neuron formation in E12.5 mutant embryos led to changes in the neuronal subtype composition of the E18.5 cortex. To this end, we used immunofluorescence labeling for Tbr1 and Ctip2 to analyze the formation of layer VI and V neurons, respectively, whereas Satb2 served as a layer II-IV marker (*Figure 4*). Inspecting these immunostainings at low magnification showed that Tbr1+, Ctip2+ and Satb2+ neurons occupied their correct relative laminar positions in *Inpp5e* mutants (*Figure 4A–F*) except for neuronal heterotopias which were present in all mutant brains, although their number and position varied (*Figure 4D*). These immunostainings also revealed a medial shift in the position of the rhinal fissure, a sulcus that is conserved across mammalian species and separates neocortex from the paleocortical piriform cortex (*Ariens-Kapers et al., 1936*). This shift was more marked caudally and suggests a dramatic expansion of the *Inpp5e* mutant piriform cortex at the expense of neocortex at caudal most levels (*Figure 4D–F*). Using the Tbr1/Ctip2 and Satb2 stainings, we determined the proportions of deep and superficial layer neurons, respectively. Because of the expanded piriform cortex in *Inpp5e* mutants, we limited this investigation to the unaffected rostral neocortex. In the rostrolateral neocortex, we found the proportion of Tbr1+

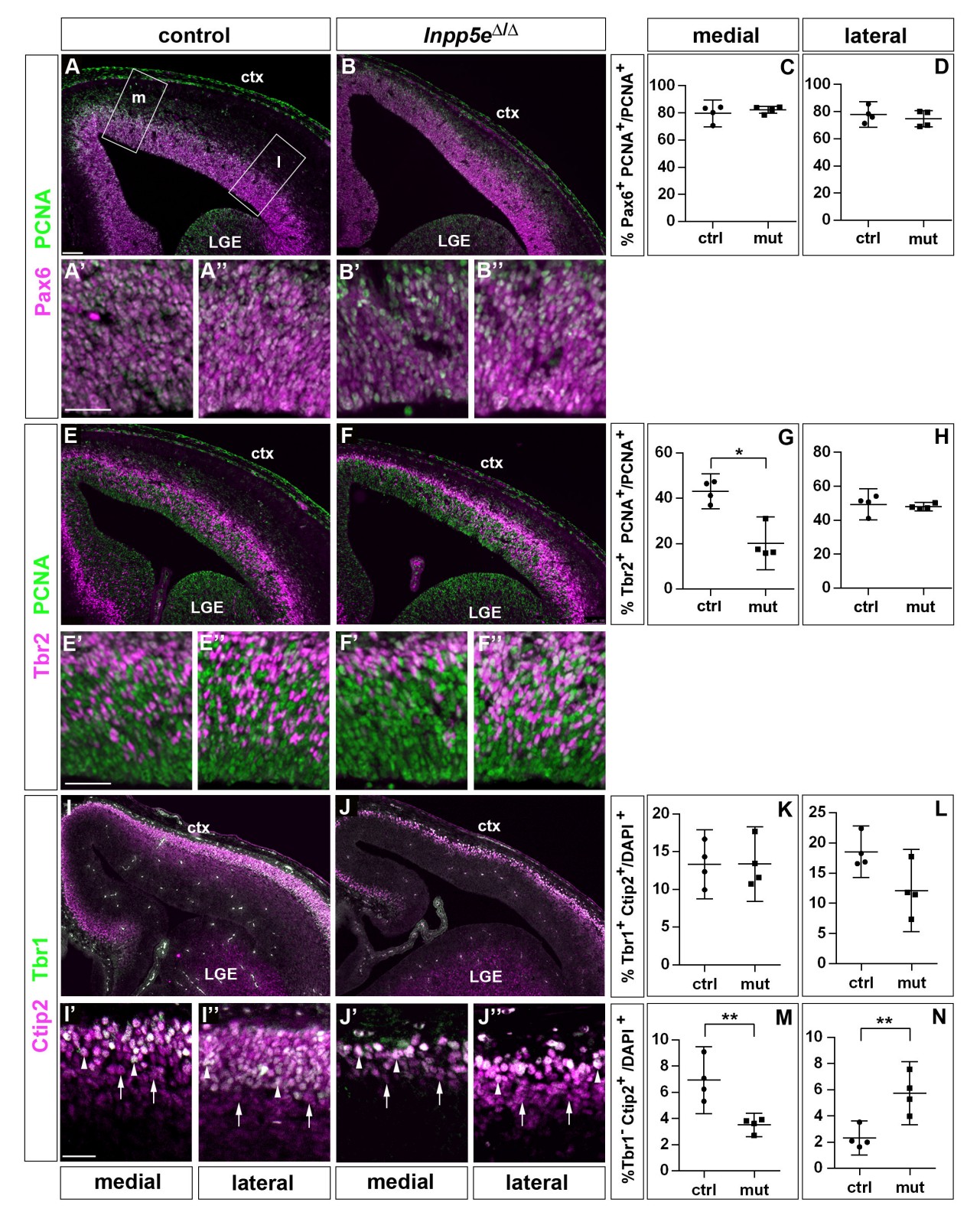

**Figure 2.** Proportions of radial glial cells, basal progenitors and neurons in the neocortex of E14.5 *Inpp5e*$^{\Delta/\Delta}$ embryos. (A–D) The proportion of radial glial cells remains unaffected by the *Inpp5e* mutation as revealed by Pax6/PCNA double immunofluorescence. The boxes in A indicate the regions in the medial (m) and lateral (l) telencephalon at which cell counts were performed. (E–H) Tbr2/PCNA double staining showed a reduced proportion of basal progenitors in the *Inpp5e*$^{\Delta/\Delta}$ medial but not lateral neocortex. (I–N) The proportion of Tbr1+Ctip2+ neurons is not significantly altered (I–L),

*Figure 2 continued on next page*

Figure 2 continued

whereas the proportion of Tbr1-Ctip2+ neurons is decreased and increased in the medial and lateral neocortex, respectively. Arrows in (I and J) label Tbr1-Ctip2+ neurons and arrowheads Tbr1+Ctip2+ neurons. (A–J) The insets labeled with ' and '' are representative magnifications of medial and lateral levels, respectively. All statistical data are presented as means ± 95% confidence intervals (CI); Unpaired t-tests (C, D, H, K–N) and Mann Whitney test (G); n = 4; *p<0.05; **p<0.01. Scale bars: 100 µm (A) and 50 µm (A', E' and I'). ctx: cortex; LGE: lateral ganglionic eminence.

neurons to be reduced (*Figure 4G,H,M*). This reduction coincided with an increased proportion of Ctip2+ layer V neurons (*Figure 4I,J,N*) while the Satb2 population was unchanged (*Figure 4K,L,O*). In contrast, the rostromedial neocortex did not show any differences (*Figure 4P–X*). Thus, the increase in direct neurogenesis in the lateral neocortex during earlier development concurs with a change in the proportions of E18.5 Tbr1+ and Ctip2+ deep layer neurons.

## A mutation in the ciliary gene *Tctn2* leads to increased telencephalic neurogenesis

To start to unravel the mechanisms by which *Inpp5e* controls cortical stem cell development, we first analyzed whether the increased early neurogenesis is restricted to *Inpp5e*$^{\Delta/\Delta}$ mutants or is observed in another mutant affecting cilia. To this end, we focused on the *Tectonic 2* (*TCTN2*) gene which is crucial for ciliary transition zone architecture (*Shi et al., 2017*) and which, like *INPP5E*, is mutated in Joubert Syndrome (*Garcia-Gonzalo et al., 2011*). Interestingly, E12.5 *Tctn2*$^{\Delta/\Delta}$ mutant embryos

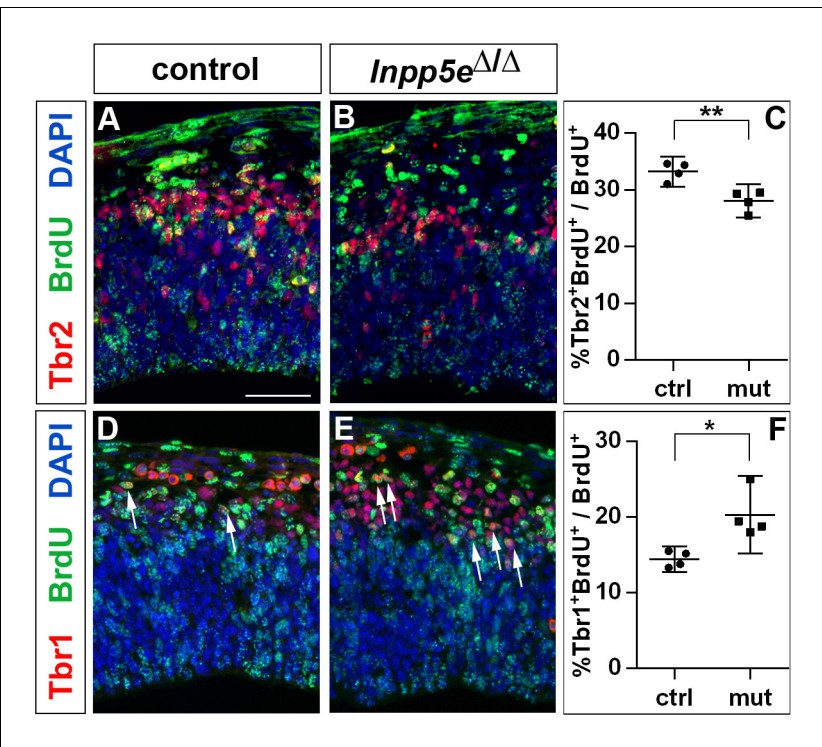

**Figure 3.** Increased neurogenesis at the expense of basal progenitor formation in the E12.5 *Inpp5e*$^{\Delta/\Delta}$ mutant lateral telencephalon. Immunohistochemistry on sections of E12.5 control (**A, D**) and *Inpp5e*$^{\Delta/\Delta}$ embryos (**B, E**) that were treated with BrdU 24 hr earlier. (**A–C**) Tbr2/BrdU double labeling showed that less basal progenitors formed from the BrdU-labeled progenitor cohort in *Inpp5e*$^{\Delta/\Delta}$ embryos. (**D–F**) The proportion of newly formed Tbr1+ neurons was increased in *Inpp5e*$^{\Delta/\Delta}$ embryos. The arrows in D and E label Tbr1$^+$BrdU$^+$ cells. All statistical data are presented as means ± 95% confidence intervals (CI); unpaired t tests; n = 4; *p<0.05; **p<0.01. Scale bar: 50 µm. The online version of this article includes the following figure supplement(s) for figure 3:

**Figure supplement 1.** Apoptosis in the developing forebrain of *Inpp5e*$^{\Delta/\Delta}$ embryos.
**Figure supplement 2.** Proportion of mitotic progenitors in *Inpp5e*$^{\Delta/\Delta}$ embryos.
**Figure supplement 3.** Cell cycle of cortical progenitors in E12.5 *Inpp5e*$^{\Delta/\Delta}$ embryos.

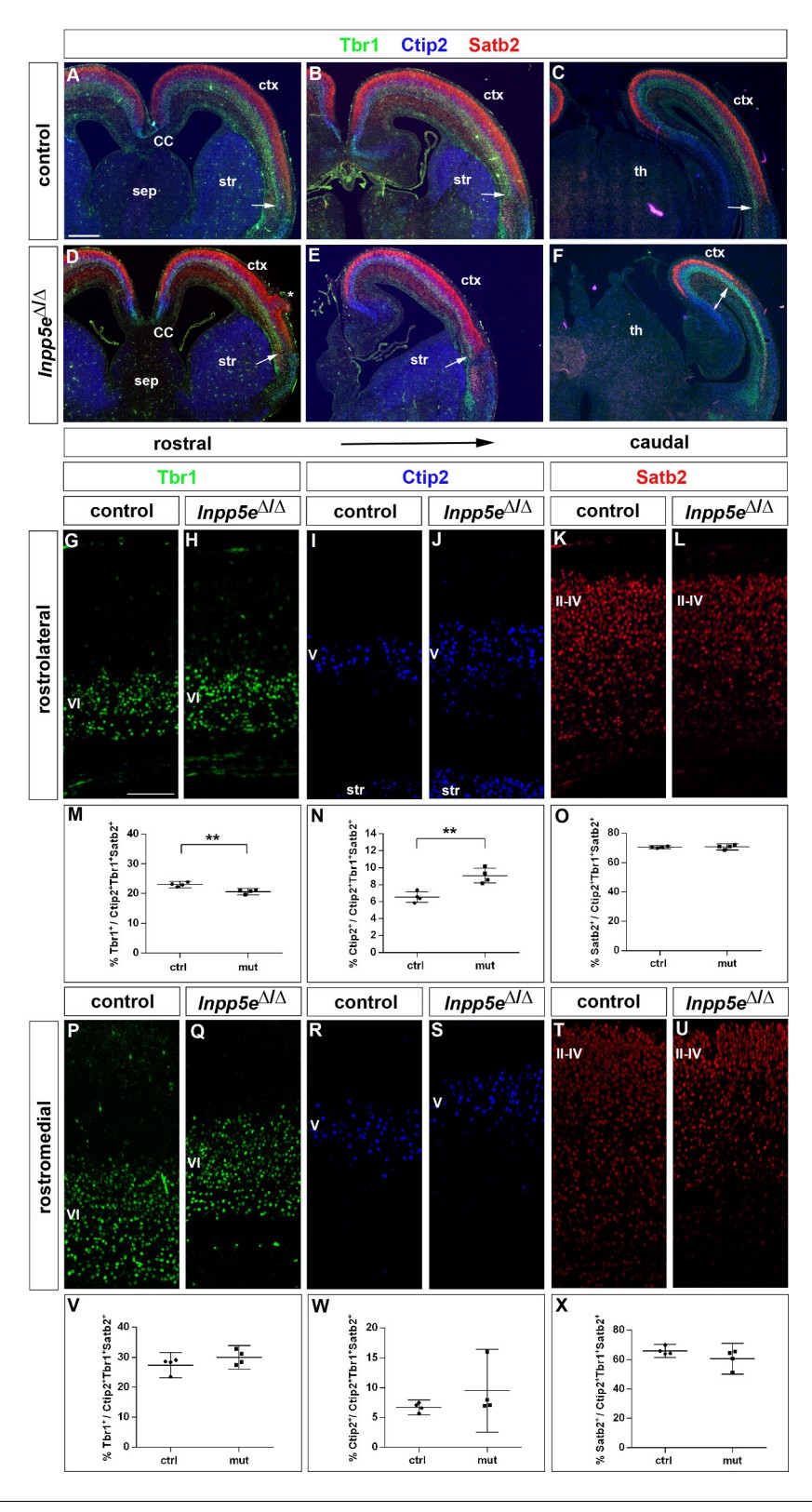

**Figure 4.** Increased formation of layer V neurons in E18.5 *Inpp5e*$^{\Delta/\Delta}$ mutants. (**A–F**) Coronal sections immunostained for the deep layer markers Tbr1 (layer VI) and Ctip2 (layer V) and for the upper layer marker Satb2 (layers II-IV); there is no obvious defect in layering in *Inpp5e*$^{\Delta/\Delta}$ embryos except for the formation of a heterotopia (asterisk in D). At caudal levels, the cortex becomes thinner and the rhinal fissure is shifted medially as indicated by the arrows. (**G–O**) Formation of cortical neurons at rostrolateral levels. The proportion of Tbr1+layer VI neurons is decreased with a concomitant increase in

*Figure 4 continued on next page*

Figure 4 continued

Ctip2+layer V neurons. (P–X) Portion of cortical neurons at rostromedial levels. Immunolabeling with cortical layer markers revealed no significant difference. Note that due to the thinner cortex, the position of layer VI Tbr1+ (**Q**) and layer V Ctip2+ neurons (**J, S**) appears to be shifted to more superficial positions; however, the relative order of these layers remains unaffected. All statistical data are presented as means ± 95% confidence intervals (CI); unpaired t-tests (**M–O, X**); Mann Whitney tests (**V, W**); n = 4; **p<0.01. Scale bars: 500 μm (**A**) and 100 μm (**G**). CC: corpus callosum; ctx: cortex; sep: septum; str: striatum.

The online version of this article includes the following figure supplement(s) for figure 4:

**Figure supplement 1.** Whole mount preparations of E18.5 brains.
**Figure supplement 2.** Forebrain malformations in E18.5 *Inpp5e*$^{\Delta/\Delta}$ embryos.
**Figure supplement 3.** Hippocampus formation in *Inpp5e*$^{\Delta/\Delta}$ embryos.
**Figure supplement 4.** Formation of the corpus callosum in E18.5 *Inpp5e*$^{\Delta/\Delta}$ embryos.

(*Reiter and Skarnes, 2006*) also showed an increased proportion of Tbr1+ projection neurons and a concomitant decrease in Tbr2+ basal progenitors in the dorsolateral telencephalon (*Figure 5*). Due to embryonic lethality, however, we were not able to investigate the formation of cortical neurons at later stages.

## Ciliary defects in the forebrain of E12.5 *Inpp5e*$^{\Delta/\Delta}$ embryos

Our findings in the *Inpp5e* and *Tctn2* mutants suggested a role for cilia in cortical progenitor cells to control early neurogenesis. Therefore, we examined the presence and the structure of primary cilia in the developing forebrain of *Inpp5e*$^{\Delta/\Delta}$ embryos by immunofluorescence and electron microscopy. We first analyzed the presence of the small GTPase Arl13b, enriched in ciliary membranes, and of γ-Tubulin (γTub), a component of basal bodies (*Caspary et al., 2007*). We found no major difference

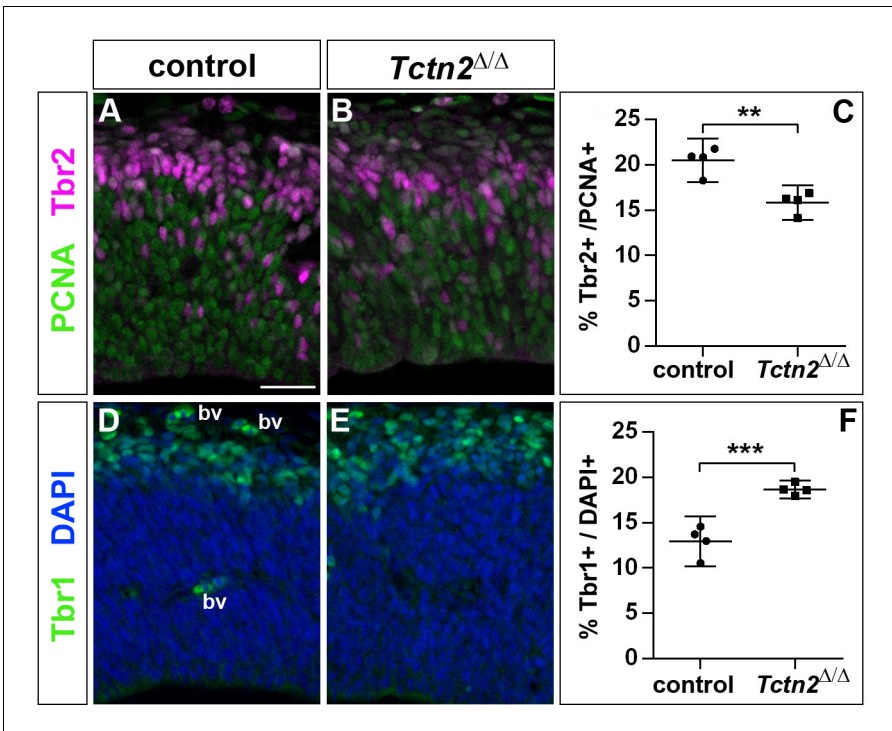

**Figure 5.** Increased generation of cortical neurons in the lateral neocortex of E12.5 *Tctn2*$^{\Delta/\Delta}$ embryos. (**A–C**) Double immunofluorescence for PCNA and Tbr2 revealed a significantly decreased proportion of basal progenitors. (**D–F**) The portion of Tbr1$^{+}$ cortical neurons was increased. All statistical data are presented as means ± 95% confidence intervals (CI); unpaired t tests; n = 4; **p<0.01; ***p<0.001. Scale bar: 50 μm. bv: blood vessel.

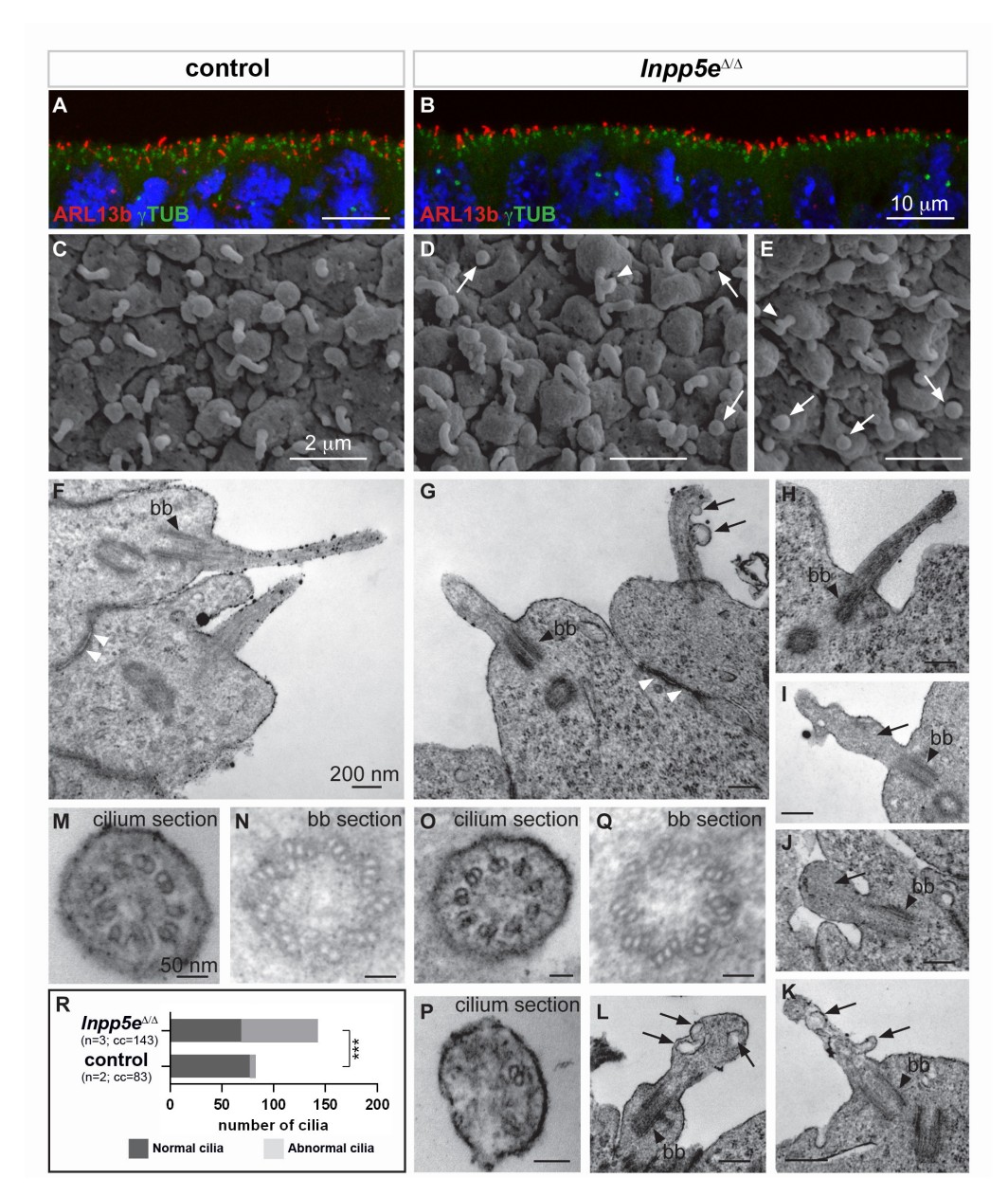

**Figure 6.** Ciliary defects in E12.5 *Inpp5e*$^{\Delta/\Delta}$ forebrain. (A–B) Immunohistochemistry for Arl13b and γ-Tubulin (γTUB) on E12.5 brain cryosections showed an accumulation of ciliary axonemes and basal bodies, respectively, at the apical border of radial glial cells facing the ventricules in both control (A) and *Inpp5e*$^{\Delta/\Delta}$ (B) embryos without any gross difference. Scale bars: 10 μm. (C–E) Scanning electron microscopy (SEM) on E12.5 control (C) and *Inpp5e*$^{\Delta/\Delta}$ (D, E) brains highlighted the presence of primary cilia projecting from the apical surface of radial glial cells in both control (A) and *Inpp5e*$^{\Delta/\Delta}$ (B) embryos. However, SEM also revealed the presence of abnormal cilia in *Inpp5e*$^{\Delta/\Delta}$ embryos having a spherical shape (arrows in D and E) or aberrant lateral buddings (arrowheads in D and E). Scale bars: 2 μm. (F–L) Transmission electron microscopy (TEM) analysis on E12.5 brains showed longitudinal sections of primary cilia in control (F) and *Inpp5e*$^{\Delta/\Delta}$ (G–L) embryos. In control primary cilia, the axoneme appeared as an extension of the basal body (bb, black arrowheads) (F–H). In addition to cilia with normal morphology, abnormal cilia were identified in *Inpp5e*$^{\Delta/\Delta}$ embryos thanks to the presence of a basal body apparently correctly docked to the apical membrane. Abnormal cilia lacked an axoneme (I, J, L) or showed unusual membranous structures, such as budding (G, K) or internal (I, K, L) vesicles (arrows) or undulating peripheral membranes (I). Note that tight junctions (white arrowheads in F and G) appeared normal in *Inpp5e*$^{\Delta/\Delta}$ (G) and control (F) embryos, suggesting that apico-basal polarity of *Inpp5e*$^{\Delta/\Delta}$radial glial cells was not compromised. Scale bars: 200 nm. (M–Q) TEM images showing transverse sections of the axoneme (M, O, P) and the basal body (N, Q) in control (M, N) and *Inpp5e*$^{\Delta/\Delta}$ (O–Q) embryos with no major difference in the basal bodies between control (N) and *Inpp5e*$^{\Delta/\Delta}$ (Q) embryos. However, transverse section of primary cilia in *Inpp5e*$^{\Delta/\Delta}$ brains revealed the presence of normal axonemes composed of nine correctly organized doublets of microtubules on some radial glial cells (O), while others harbored an abnormal axoneme containing a lower number of microtubule doublets (P). Scale

*Figure 6 continued on next page*

*Figure 6 continued*

bars: 50 nm. (**R**) Graph showing the number of normal versus abnormal cilia (cil.) found on TEM images from control (n = 3) or *Inpp5e*$^{\Delta/\Delta}$ (n = 3) embryos. cc: counted cilia.

in the number or the apical localization of cilia in control and *Inpp5e*$^{\Delta/\Delta}$ neuroepithelial cells in the E12.5 telencephalon (*Figure 6A,B*) or diencephalon (data not shown).

To gain insights into the fine structure of these primary cilia, we performed electron microscopy analyses. Scanning electron microscopy (SEM) provided an observation of the cilia protruding into the telencephalic ventricles. In control embryos, almost all RGCs had a single, ~1 μm long primary cilium (*Figure 6C*), as previously described (*Besse et al., 2011*). Some *Inpp5e*$^{\Delta/\Delta}$ mutant cells also displayed an apparently normal cilium (*Figure 6D,E*), whereas other cells harbored abnormal cilia, either with a lateral blob (arrowhead in *Figure 6D,E*) or as a short and bloated cilium-like protrusion (arrows in *Figure 6D,E*).

Transmission electron microscopy (TEM) confirmed the presence of abnormal cilia in *Inpp5e*$^{\Delta/\Delta}$ embryos. Cilia were recognized by basal bodies anchored to the apical membrane in both control and *Inpp5e*$^{\Delta/\Delta}$RGCs (*Figure 6F–L,N,Q*). However, in *Inpp5e*$^{\Delta/\Delta}$ cells, some cilia lacked the axoneme and showed unusual membranous structures that resemble budding vesicles emerging from the lateral surface of the cilium (*Figure 6G,K*), internal vesicles (arrows in *Figure 6I,K,L*), or undulating peripheral membranes (*Figure 6I*), indicating an *Inpp5e*-dependent defect in ciliary membrane morphology. Transverse sections revealed the presence of cilia with apparently normal 9+0 axonemes, as well as cilia containing abnormal numbers of microtubule doublets in *Inpp5e*$^{\Delta/\Delta}$ embryos (*Figure 6O,P*). To quantify these ciliary defects, we counted the number of normal versus abnormal cilia on TEM images obtained from control and *Inpp5e*$^{\Delta/\Delta}$ embryos, and found an increase in abnormal cilia in *Inpp5e*$^{\Delta/\Delta}$ compared to control embryos (*Figure 6R*). Taken together, a significant number of abnormal primary cilia were found at the apical end of E12.5 RGCs in the forebrain of *Inpp5e*$^{\Delta/\Delta}$ embryos. These abnormalities are consistent with a role of *Inpp5e* in maintaining cilia stability (*Jacoby et al., 2009*).

## Restoring Gli3 repressor ratio rescues cortical malformations in *Inpp5e*$^{\Delta/\Delta}$ embryos

Primary cilia play a crucial role in Shh signaling by controlling the proteolytic cleavage of full-length Gli3 (Gli3FL) into the Gli3 repressor form (Gli3R) in the absence of Shh and by converting Gli3FL into the transcriptional activator Gli3A in the presence of Shh. Moreover, the dorsal telencephalon predominately forms Gli3R (*Fotaki et al., 2006*) and mice that can only produce Gli3R have no obvious defect in cortical development (*Besse et al., 2011*; *Bose, 2002*). In addition, we recently showed that Gli3 has a prominent role in RGCs controlling the switch from symmetric proliferative to asymmetric neurogenic cell division (*Hasenpusch-Theil et al., 2018*). Therefore, we hypothesized that alterations in Gli3 processing caused by abnormal cilia function underlies the increased direct neurogenesis and the cortical malformations in *Inpp5e*$^{\Delta/\Delta}$ embryos. In situ hybridization showed that *Gli3* mRNA expression might be slightly reduced but the overall expression pattern in the telencephalon remains unaffected (*Figure 7—figure supplement 1*). We next investigated the formation of Gli3FL and Gli3R in the E12.5 dorsal telencephalon of *Inpp5e*$^{\Delta/\Delta}$ embryos using western blots. This analysis revealed no change in the levels of Gli3FL but a significant decrease inGli3R which resulted in a reduced Gli3R to Gli3FL ratio in the mutant (*Figure 7A–D*) suggesting that the *Inpp5e* mutation affects Gli3 processing.

The next set of experiments aimed to clarify a role for the reduced Gli3 processing. To this end, we restored Gli3R levels by crossing *Inpp5e* mutants with *Gli3*$^{\Delta699/+}$ mice that can only produce Gli3R in a cilia-independent manner (*Besse et al., 2011*; *Bose, 2002*). Overall inspection of *Inpp5e*$^{\Delta/\Delta}$;*Gli3*$^{\Delta699/+}$ embryos revealed restored eye formation, whereas *Inpp5e*$^{\Delta/\Delta}$ embryos either completely lacked eyes or showed microphthalmia (*Jacoby et al., 2009*; *Figure 7—figure supplement 2*). Moreover, the overall morphology of the telencephalon is much improved in *Inpp5e*$^{\Delta/\Delta}$;*Gli3*$^{\Delta699/+}$ embryos as compared to *Inpp5e*$^{\Delta/\Delta}$ embryos. In E18.5 *Inpp5e*$^{\Delta/\Delta}$*Gli3*$^{\Delta699/+}$ mutants, the corpus callosum has a thickness indistinguishable from that of control embryos (*Figure 7—figure supplement 3*). In E12.5 and E14.5 *Inpp5e*$^{\Delta/\Delta}$;*Gli3*$^{\Delta699/+}$ embryos, the neocortex lacks the undulations of the VZ

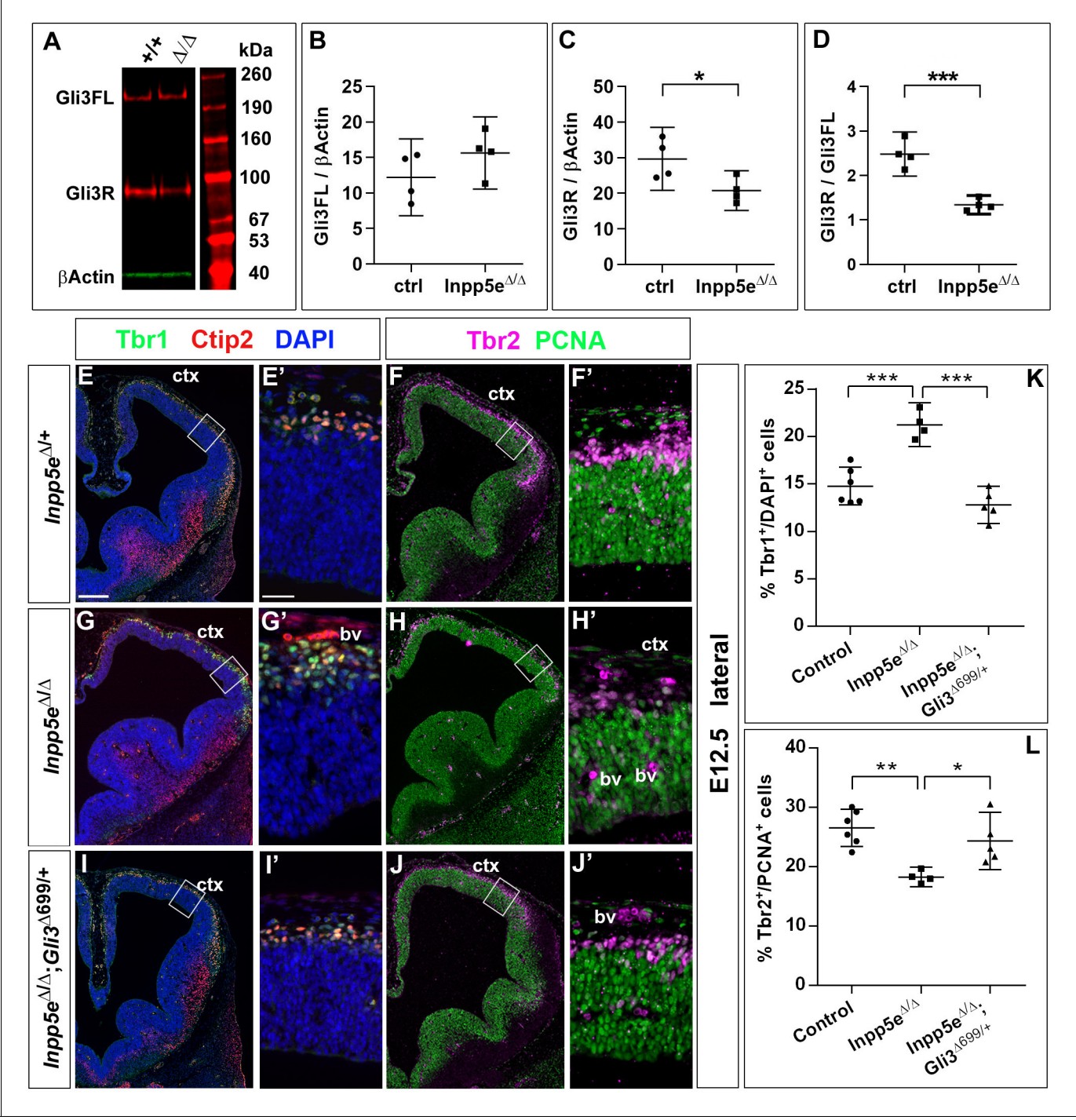

**Figure 7.** Re-introducing a single copy of the Gli3 repressor rescues the neurogenesis defect in E12.5 *Inpp5e* mutants. (A–D) Gli3 western blot on E12.5 dorsal telencephalic tissue revealed the Gli3 full length (FL) and repressor (R) forms (A). While Gli3FL levels are not affected (B), levels of Gli3R (C) and the Gli3R/Gli3FL ratio (D) are decreased in *Inpp5e*$^{\Delta/\Delta}$ embryos. An unpaired t-test was used to evaluate levels of Gli3FL and Gli3R and the Gli3R/Gli3FL ratio in four control and four*Inpp5e*$^{\Delta/\Delta}$ embryos derived from four different litters. (E–L) Formation of basal progenitors and neurons in the neocortex of *Inpp5e*$^{\Delta/\Delta}$ and *Inpp5e*$^{\Delta/\Delta}$;Gli3$^{\Delta699/+}$ embryos. In the lateral neocortex of E12.5 embryos, there is no significant difference in the proportions of Tbr1+ neurons (E, G, I, K) and basal progenitor cells (F, H, J, L) between control and *Inpp5e*$^{\Delta/\Delta}$;Gli3$^{\Delta699/+}$ embryos. Note the three bulges of the ventral telencephalon in *Inpp5e*$^{\Delta/\Delta}$;Gli3$^{\Delta699/+}$ embryos (J). Boxes indicate the regions where cell counts were performed. All statistical data are presented as means ± 95% confidence intervals (CI); unpaired t-tests (n = 5) (B–D) and one-way ANOVA followed by Tukey's multiple comparison test (K, L); *p<0.05; **p<0.01; ***p<0.001. Scale bars: 250 µm (E), and 50 µm (E'). bv: blood vessel; ctx: cortex.

The online version of this article includes the following figure supplement(s) for figure 7:

*Figure 7 continued on next page*

Figure 7 continued

**Figure supplement 1.** *Gli3* mRNA expression in Inpp5e mutants.
**Figure supplement 2.** Rescue of eye development in *Inpp5e*$^{\Delta/\Delta}$;*Gli3*$^{\Delta699/+}$ embryos.
**Figure supplement 3.** Rescue of corpus callosum formation in *Inpp5e*$^{\Delta/\Delta}$;*Gli3*$^{\Delta699/+}$ embryos.

present in *Inpp5e*$^{\Delta/\Delta}$ embryos (data not shown) and the morphology of the hippocampal anlage is more akin to that in wild-type embryos but it is still smaller and less bulged (*Figure 7E,G,I*).

We also determined the proportions of basal progenitors and Tbr1+ neurons at E12.5 which were decreased and increased, respectively, in the lateral neocortex of *Inpp5e*$^{\Delta/\Delta}$ embryos. While these changes were still present in *Inpp5e*$^{\Delta/\Delta}$ littermate embryos, there was no statistically significant difference between control and *Inpp5e*$^{\Delta/\Delta}$;*Gli3*$^{\Delta699/+}$ embryos (*Figure 7E–L*). This finding indicates that the neurogenesis phenotype of E12.5 *Inpp5e*$^{\Delta/\Delta}$ mutants is rescued by a single copy of the Gli3$^{\Delta699}$ allele. We next investigated the formation of basal progenitors and of cortical projection neurons in E14.5 *Inpp5e*$^{\Delta/\Delta}$;*Gli3*$^{\Delta699/+}$ embryos. The proportion of Tbr1+Ctip2+ neurons was not affected in the medial neocortex of E14.5 *Inpp5e*$^{\Delta/\Delta}$;*Gli3*$^{\Delta699/+}$ embryos. In contrast, the proportion of Tbr1-Ctip2+ neurons was reduced as in *Inpp5e*$^{\Delta/\Delta}$ mutants (*Figure 8A,C,E,I,J*). Similarly, the proportions of basal progenitors in the medial *Inpp5e*$^{\Delta/\Delta}$*Gli3*$^{\Delta699/+}$ neocortex was slightly improved compared to *Inpp5e*$^{\Delta/\Delta}$ embryos but significantly smaller than in control embryos (*Figure 8B,D,F,K*). As re-introducing a single Gli3$^{\Delta699}$ allele does not completely rescue the *Inpp5e*$^{\Delta/\Delta}$ neurogenesis phenotype, we generated *Inpp5e*$^{\Delta/\Delta}$ embryos homozygous for the *Gli3*$^{\Delta699}$ allele. Interestingly, the morphology of the dorsal telencephalon including the hippocampal formation was indistinguishable between control and *Inpp5e*$^{\Delta/\Delta}$;*Gli3*$^{\Delta699/\Delta699}$ embryos (*Figure 8A,B,G,H*) and the formation of Tbr1-Ctip2+ neurons and Tbr2+ basal progenitors were not affected (*Figure 8G,H,J,K*). Taken together, these findings indicate that re-introducing a single copy of the *Gli3R* allele into the *Inpp5e* mutant background leads to a partial rescue of cortical neurogenesis in *Inpp5e*$^{\Delta/\Delta}$ embryos, whereas two copies are required for a full rescue.

## Discussion

Generating a functional cerebral cortex requires a finely tuned balance between direct and indirect neurogenesis to form subtypes of cortical projection neurons in appropriate numbers. Here, we show that the ciliary mouse mutants *Inpp5e* and *Tctn2* present with a transient increase in neurons forming directly from radial glia progenitors in the lateral neocortex at the expense of basal progenitor formation. This increase in neurogenesis results in augmented formation of Ctip2+ layer V neurons in the *Inpp5e* mutant cortex. Our studies also revealed that the *Inpp5e* mutation interfered with the stability of the RGC primary cilium and its signaling functions, leading to a reduction in the Gli3R levels. Since re-introducing Gli3R in an *Inpp5e* mutant background restored the decreased formation of normal proportions of basal progenitors and neurons, our findings implicate a novel role for primary cilia in controlling the signaling events that direct the decision of RGCs to undergo either direct or indirect neurogenesis.

### Primary cilia affect the decision between direct and indirect neurogenesis

RGCs in the developing mouse neocortex have the potential to undergo symmetric proliferative or asymmetric cell divisions with the latter division mode producing neurons in a direct manner or indirectly via basal progenitors. Balancing out these division modes is important not only to determine final neuronal output and cortical size but also the types of cortical projection neurons and, hence, subtype composition of the adult neocortex. In the E12.5 *Inpp5e* and *Tctn2* mouse mutants, we identified an increased formation of neurons in the lateral neocortex. Based on our cell cycle exit experiment, additional neurons are formed from RGCs at the expense of basal progenitors. Given the cell cycle length of basal progenitors of >24 hr (*Arai et al., 2011*), it is unlikely that new born basal progenitors would have undergone an additional round of cell division to produce two neurons within the time frame of this experiment. Such an extra division would also have diluted the BrdU label. We therefore conclude that the *Inpp5e* mutation caused RGCs to preferentially produce

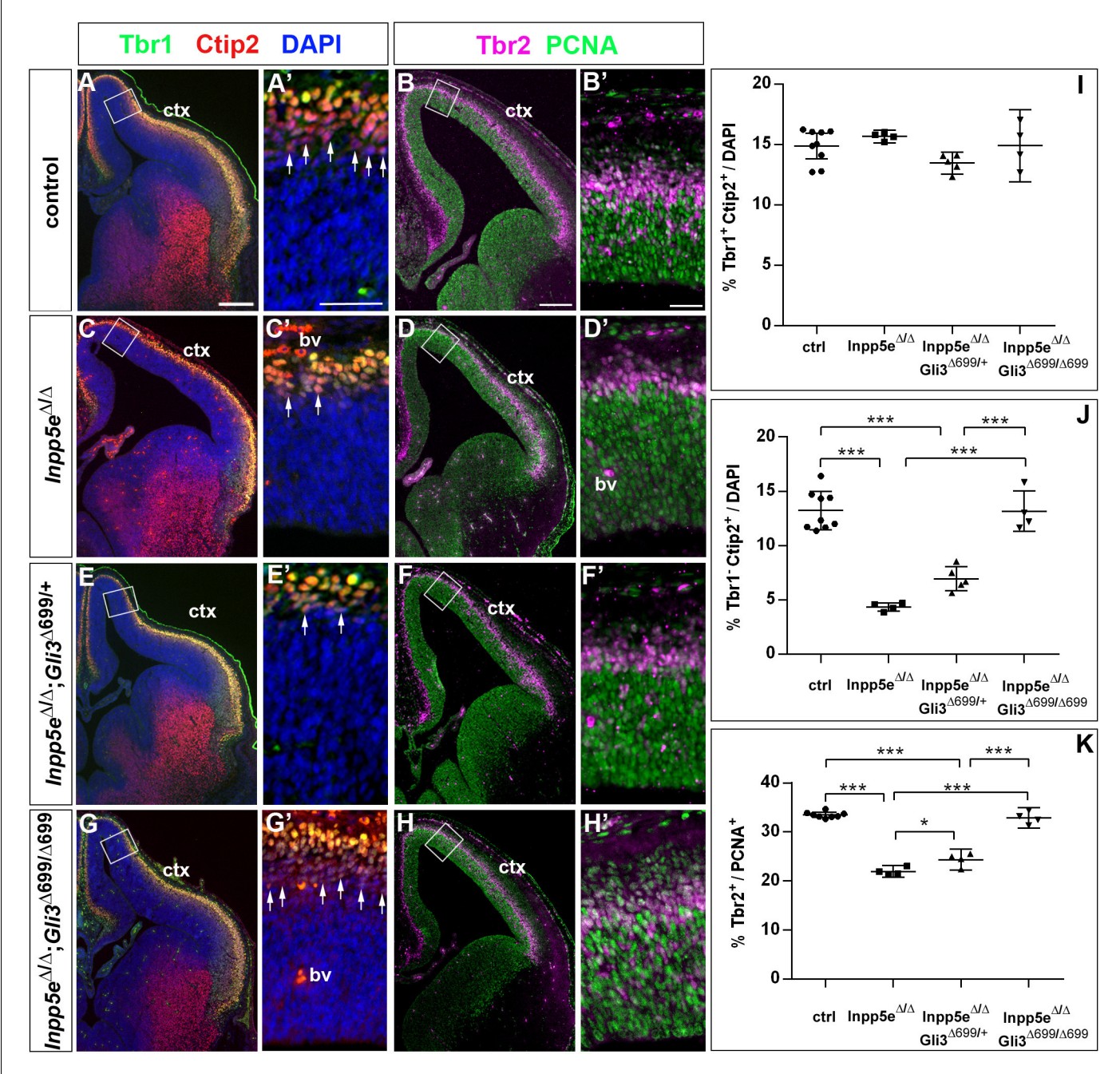

**Figure 8.** Two copies of the *Gli3* repressor allele are required to rescue the neurogenesis defects in E14.5 *Inpp5e* mutants. (A–H) Proportions of neurons (A, C, E, G) and basal progenitors (B D, F, H) in the medial neocortex of control, *Inpp5e*$^{\Delta/\Delta}$, *Inpp5e*$^{\Delta/\Delta}$;Gli3$^{\Delta699/+}$ and *Inpp5e*$^{\Delta/\Delta}$;Gli3$^{\Delta699/\Delta699}$ embryos. (A, C, E, G, J) The formation of Tbr1-Ctip2+ projection neurons is rescued after re-introducing two copies of the *Gli3* repressor allele. (B, D, F, H, K) The proportion of basal progenitors is slightly increased in *Inpp5e*$^{\Delta/\Delta}$;Gli3$^{\Delta699/+}$ embryos but a full rescue is only achieved in *Inpp5e*$^{\Delta/\Delta}$;Gli3$^{\Delta699/\Delta699}$ embryos. Boxes indicate the regions where cell counts were performed. All statistical data are presented as means ± 95% confidence intervals (CI); one-way ANOVA followed by Tukey's multiple comparison test (I, J, K); *p<0.05; ***p<0.001. Scale bars: 250 µm (A, B), 50 µm (A', B'). bv: blood vessel; ctx: cortex.

neurons directly. Moreover, neurogenesis defects only became obvious at E14.5 in the medial neocortex. This delay might reflect the neurogenic gradient in the neocortex or might be related to specific gene expression changes such as reduced Pax6 expression in medial neocortical progenitors.

Interestingly, the increase in direct neurogenesis led to an increased proportion of Ctip2+ deep layer V neurons in the E18.5 neocortex but did not coincide with a reduced proportion of upper layer neurons. This effect could be explained in several mutually non-exclusive ways. First, neurons born at E12.5 initially express both Ctip2 and Tbr1 (*Figure 7*) and later down-regulate Ctip2. *Inpp5e* could therefore affect the signaling that controls this downregulation. Secondly, the proportions of basal progenitors and neurons were normalized in E14.5 mutants. Since basal progenitors are a main source of upper layer neurons (*Arnold et al., 2008*; *Vasistha et al., 2015*), this normalization would account for the sufficient numbers of Satb2+ upper layer neurons. Newly formed projection neurons signal back to RGCs via Jag1, Fgf9 and Neurotrophin 3 (*Parthasarathy et al., 2014*; *Seuntjens et al., 2009*; *Wang et al., 2016*) to control the sequential production of deep and upper layer neurons and of glia (*Silva et al., 2019*). *Inpp5e* might affect these signals by controlling cilia stability and/or levels of PI(3,4,5)P$_3$ (*Bielas et al., 2009*; *Jacoby et al., 2009*) that acts as a second messenger in receptor tyrosine kinase signaling. Regardless of the exact mechanism, our findings suggest a novel, spatially and temporally restricted role for *Inpp5e* in controlling the decision between direct and indirect neurogenesis. This function differs from those described for other cilia mutants. Conditional inactivation of *Ift88* and *Kif3a* leads to a larger cortex (*Foerster et al., 2017*; *Wilson et al., 2012*) with a modest increase in BP production in the absence of a delay in neurogenesis (*Foerster et al., 2017*) while *Rpgrip1l* mutants have reduced numbers of both basal progenitors and neurons (*Postel et al., 2019*). These findings highlight the multiple and varied roles cilia play in cortical development.

### *Inpp5e* controls direct/indirect neurogenesis through Gli3 processing

Our study also shed lights into the mechanisms by which *Inpp5e* controls the decision between direct and indirect neurogenesis. Most notably, the Gli3R level and Gli3R/Gli3FL ratio are decreased in *Inpp5e*$^{\Delta/\Delta}$ embryos. While the *Inpp5e* mutation does not lead to an up-regulation of Shh signaling in the dorsal telencephalon (*Magnani et al., 2015*), re-introducing a single or two copies of Gli3R in an *Inpp5e* mutant background partially and fully restores the neurogenesis defects, respectively. This rescue indicates that reduced levels of Gli3R rather than the reduction in the Gli3R/Gli3FL ratio are responsible for the prevalence of direct neurogenesis in *Inpp5e*$^{\Delta/\Delta}$ embryos. This idea is consistent with the findings that (i) *Gli3*$^{\Delta699/\Delta699}$ embryos that cannot produce Gli3FL and Gli3A show no obvious phenotype in cortical development (*Besse et al., 2011*; *Bose, 2002*), (ii) dorsal telencephalic patterning defects in *Gli3*$^{Xt/Xt}$ mutants are not rescued in *Shh*$^{-/-}$/*Gli3*$^{XtXt}$ double mutants (*Rallu et al., 2002*; *Rash and Grove, 2007*), (iii) Shh promotes the generation of olfactory bulb interneurons and cortical oligodendrocytes and neurogenesis in the subventricular zone by reducing Gli3R rather than by promoting Gli activator function (*Petrova et al., 2013*; *Wang et al., 2014*; *Zhang et al., 2020*). In addition, there is also a dramatic rescue of eye development and the rescue also extends to other malformations of the *Inpp5e*$^{\Delta/\Delta}$ forebrain, including the corpus callosum, the hippocampus and the expansion of the piriform cortex, structures that are also affected in *Gli3* null and hypomorphic mutants (*Amaniti et al., 2015*; *Johnson, 1967*; *Magnani et al., 2014*; *Theil et al., 1999*; *Wiegering et al., 2019*). Taken together, these findings support the idea that *Inpp5e* and the primary cilium control key processes in cortical development by regulating the formation of Gli3R.

Our analyses support several mutually non-exclusive mechanisms how the *Inpp5e* mutation impacts on Gli3 processing. First, our electron microscopy study revealed severe structural abnormalities in large proportions of cilia. The Inpp5e phosphatase hydrolyses PI(3,4,5)P$_3$, which is essential for the effective activation of the serine threonine kinase Akt (*Kisseleva et al., 2002*; *Plotnikova et al., 2015*). Following PI(3,4,5)P$_3$ binding, Akt translocates to the membrane and becomes phosphorylated at T308 by phosphoinositide-dependent kinase-1 (Pdk1) and at S473 by mammalian target of rapamycin complex (mTORC2) (*Yu and Cui, 2016*). Consistent with the loss of *Inpp5e* function and a resulting increase in PI(3,4,5)P$_3$, western blot analysis revealed elevated pAkt$^{S473}$ levels (data not shown). Increased phosphorylation at this site has been implicated in inhibiting cilia assembly and promoting cilia disassembly (*Mao et al., 2019*) and could hence explain the structural defects of RGC *Inpp5e*$^{\Delta/\Delta}$ cilia. Secondly, *Inpp5e* could control Gli3 processing through its effect on the transition zone (TZ). It is required for TZ molecular organization (*Dyson et al., 2017*) and its substrate PI(4,5)P2 plays a role in TZ maturation in *Drosophila* (*Gupta et al., 2018*). This model is further supported by our finding that a mouse mutant for the TZ protein Tctn2 phenocopies the *Inpp5e*$^{\Delta/\Delta}$ neurogenesis defect. In turn, several mouse mutants defective for TZ proteins are

required for Inpp5e localization to cilia and show microphthalmia (*Garcia-Gonzalo et al., 2011*; *Garcia-Gonzalo et al., 2015*; *Sang et al., 2011*; *Yee et al., 2015*). Tctn proteins are also required for Gli3 processing (*Garcia-Gonzalo et al., 2011*; *Sang et al., 2011*; *Thomas et al., 2012*; *Wang et al., 2017*) and the TZ protein Rpgrip1l controls the activity of the proteasome at the basal body responsible for proteolytic cleavage of Gli3 (*Gerhardt et al., 2015*). Taken together, these findings indicate that *Inpp5e* mutation might affect the ability of RGCs to switch to indirect neurogenesis through defects in cilia stability and/or the integrity of the ciliary transition zone (*Figure 9*).

## Implications for Joubert syndrome

In humans, hypomorphic *INPP5E* mutations contribute to Joubert Syndrome (JS), a ciliopathy characterized by cerebellar malformations and concomitant ataxia and breathing abnormalities. In addition, a subset of JS patients exhibit cortical abnormalities including polymicrogyria, neuronal heterotopias and agenesis of the corpus callosum (*Poretti et al., 2011*). Strikingly, the *Inpp5e* mouse mutant also shows several of these abnormalities. In the caudal telencephalon, the otherwise lissencephalic cortex formed folds reminiscent of the polymicrogyria in JS patients. In addition, the mutant formed

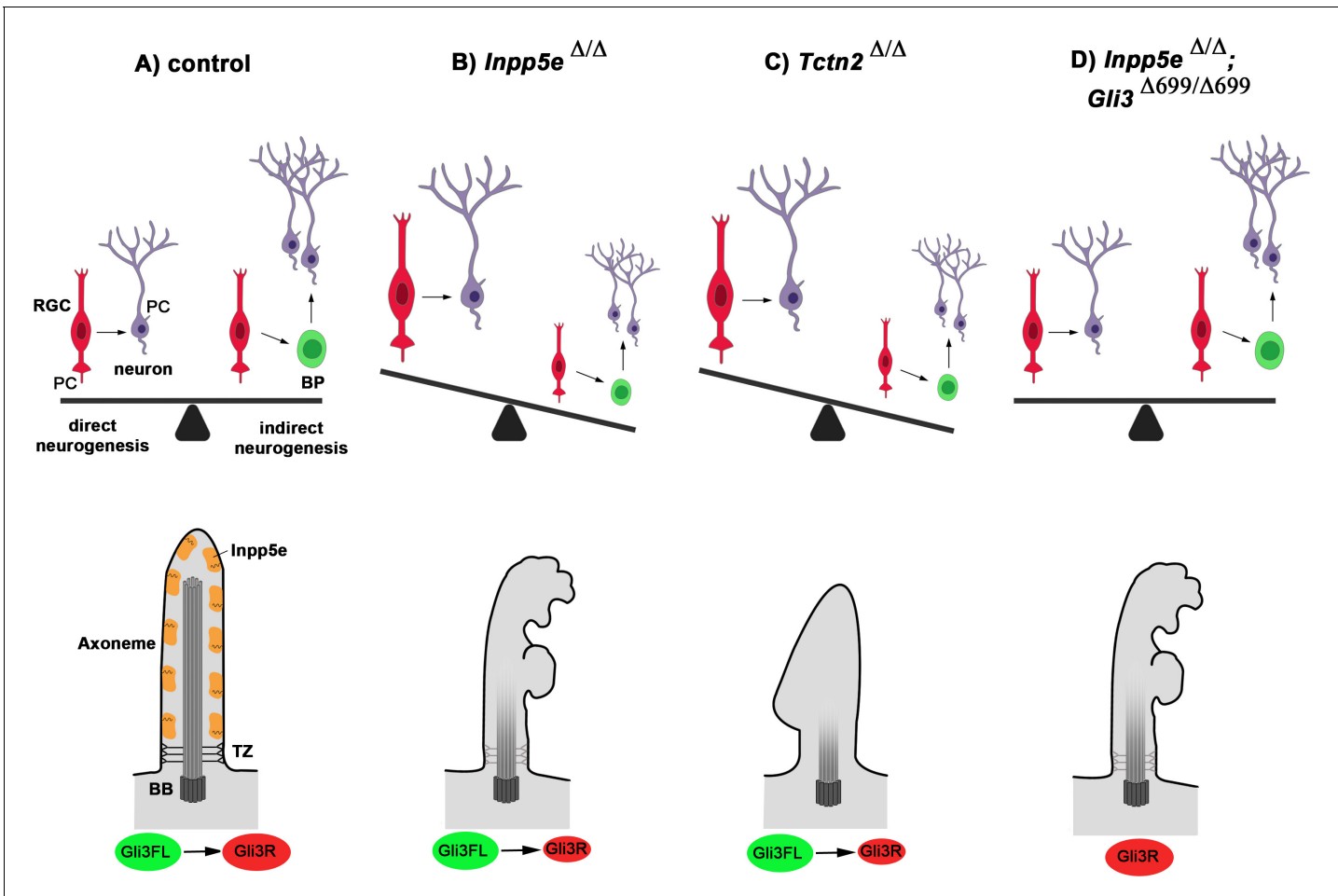

**Figure 9.** Model for *Inpp5e*'s role in controlling direct vs indirect neurogenesis in the developing cortex. (A) A fine-tuned balance between direct and indirect neurogenesis is required to produce cortical neurons in appropriate numbers. The structure of a primary cilium and the ciliary localization of the Inpp5e protein are schematically indicated. (B) The *Inpp5e* mutation affects the axoneme (shaded microtubules) and ciliary morphology and may compromise the transition zone as indicated by the grayish colour (*Dyson et al., 2017*). Gli3R levels are reduced and there is a shift towarddirect neurogenesis. (C) *Tctn2*^Δ/Δ embryos have morphologically abnormal cilia, a defective axoneme and transition zone (*Garcia-Gonzalo et al., 2011*), lack ciliary Inpp5e protein (*Garcia-Gonzalo et al., 2015*) and phenocopy the neurogenesis defect of *Inpp5e*^Δ/Δ mutants. (D) Introducing Gli3R in an *Inpp5e* mutant background restores Gli3 levels and the balance between direct and indirect neurogenesis. BB: basal body; BP: basal progenitor; PC: primary cilium; RGC: radial glial cell; TZ: transition zone.

leptomeningeal heterotopias with 100% penetrance, but their number and location varied. Mutations in ciliary genes were previously associated with heterotopia formation in humans and mice (*Magnani et al., 2015*; *Uzquiano et al., 2019*). Mice carrying mutations in the *Eml1* gene encoding a microtubule-associated protein show subcortical heterotopias due to a mispositioning of RGCs and impaired primary cilia formation (*Uzquiano et al., 2019*). Finally, the corpus callosum is thinner but callosal axons project to the contralateral cerebral hemisphere in *Inpp5e* mutants. This phenotype is milder compared to that of other mouse mutants with altered cilia that show complete agenesis of the corpus callosum with callosal axons forming Probst bundles (*Benadiba et al., 2012*; *Laclef et al., 2015*; *Putoux et al., 2019*). Unlike these other ciliary mutants, the corticoseptal boundary which plays a crucial role in positioning guidepost cells that control midline crossing of callosal axons (*Magnani et al., 2014*) is not obviously affected in *Inpp5e*$^{\Delta/\Delta}$ embryos. Instead, the thinner corpus callosum is likely to be the result of reduced size of the caudal neocortex. Despite these differences, however, re-introducing Gli3R into the cilia mutant background restores callosal development in both groups of mutants suggesting that cilia control two independent steps in corpus callosum formation by regulating Gli3 processing. Thus, the *Inpp5e*$^{\Delta/\Delta}$ mutant recapitulates cortical abnormalities in JS patients and starts to help unravelling the pathomechanisms underlying these defects.

# Materials and methods

## Key resources table

| Reagent type (species) or resource | Designation | Source or reference | Identifiers | Additional information |
|---|---|---|---|---|
| Genetic reagent (*Mus musculus*) | Inpp5e$^{delta}$ (Inpp5e$^{tm1.2Sch}$) | PMID:19668215 | MGI:4360187 | |
| Genetic reagent (*Mus musculus*) | Gli3$^{delta699}$ (Gli3$^{tm1Urt}$) | PMID:11978771 | MGI:2182576 | |
| Genetic reagent (*Mus musculus*) | Tctn2$^{delta}$ (Tctn2$^{tm1.1Reit}$) | PMID:21725307 | MGI:5292130 | |
| Antibody | Anti-Arl13b (clone N295B/66) (Mouse monoclonal) | UC Davis/NIH NeuroMab Facility | Cat# 75–287 RRID:AB_11000053 | IF (1:1500) |
| Antibody | Anti-BrdU (Rat monoclonal) | Abcam | Cat# ab6326 RRID:AB_305426 | IF (1:50) |
| Antibody | Anti-BrdU/IdU (B44) (Mouse monoclonal) | BD Biosciences | Cat# 347580 RRID:AB_2313824 | IF (1:500) |
| Antibody | Cleaved-Caspase3 (Asp175) (5A1E) (Rabbit polyclonal) | Cell Signaling Technology | Cat# 9664 RRID:AB_2070042 | IF (1:100) |
| Antibody | Anti-Ctip2 (Rat monoclonal) | Abcam | Cat# ab18465 RRID:AB_2064130 | IF (1:1000) |
| Antibody | Anti-GFAP (Rabbit polyclonal) | Agilent | Cat# Z0334 RRID:AB_10013382 | IF (1:1000) |
| Antibody | Anti-L1, clone 324 (Rat monoclonal) | Millipore | Cat# MAB5272 RRID:AB_2133200 | IF (1:1000) |
| Antibody | Anti-Pax6 (Rabbit polyclonal) | Biolegend | Cat# 901301 RRID:AB_2565003 | IF (1:400) |
| Antibody | Anti-PCNA (PC10) (Mouse monoclonal) | Abcam | Cat# ab29 RRID:AB_303394 | IF (1:500) |
| Antibody | Anti-Prox1 (Rabbit polyclonal) | Reliatech | Cat# 102-PA32 RRID:AB_10013821 | IHC (1:1000) |

*Continued on next page*

*Continued*

| Reagent type (species) or resource | Designation | Source or reference | Identifiers | Additional information |
|---|---|---|---|---|
| Antibody | Anti-pHH3 (Rabbit polyclonal) | Millipore | Cat# 06–570 RRID:AB_310177 | IF (1:100) |
| Antibody | Anti-Satb2 (Mouse monoclonal) | Abcam | Cat# ab51502 RRID:AB_882455 | IF (1:200) |
| Antibody | Anti-Tbr1 (Rabbit polyclonal) | Abcam | Cat# ab31940 RRID:AB_2200219 | IF (1:400) |
| Antibody | Anti-Tbr2 (Rabbit polyclonal) | Abcam | Cat# ab23345 RRID:AB_778267 | IF (1:1000) |
| Antibody | Anti−γTUB, (Rabbit polyclonal) | Sigma Aldrich | Cat# SAB4503045 RRID:AB_10747615 | IF (1:100) |
| Antibody | Anti-mouse Cy2 secondary (Donkey polyclonal) | Jackson ImmunoResearch Labs | Cat# 715-225-151 RRID:AB_2340827 | IF (1:100) |
| Antibody | Anti-rabbit Cy3 secondary (Donkey polyclonal) | Jackson ImmunoResearch Labs | Cat# 711-165-152 | IF (1:100) |
| Antibody | Anti-rat Cy3 secondary (Goat polyclonal) | Jackson ImmunoResearch Labs | Cat# 711-165-152 RRID:AB_2307443 | IF (1:100) |
| Antibody | Anti-rabbit Alexa Fluor 488 secondary (Goat polyclonal) | Molecular Probes (now: Invitrogen) | Cat# A-11008 RRID:AB_143165 | IF (1:200) |
| Antibody | Anti-rat Alexa Fluor 647 secondary (Goat polyclonal) | Molecular Probes (now: Invitrogen) | Cat# A-21247 RRID:AB_141778 | IF (1:200) |
| Antibody | Biotinylated swine anti-rabbit IgG | Dako | Cat# E0431 | IF (1:400) |
| Antibody | Streptavidin, Alexa Fluor 488 conjugate antibody | Molecular Probes (now: Invitrogen) | Cat# S32354 RRID:AB_2315383 | IF (1:100) |
| Antibody | Streptavidin, Alexa Fluor 568 conjugate antibody | Thermo Fisher Scientific | Cat# S-11226 RRID:AB_2315774 | IF (1:100) |
| Antibody | Biotinylated goat anti-rabbit IgG | Dako (now: Agilent) | Cat# E0432 RRID:AB_2313609 | IF (1:400) |
| Antibody | Anti-h/m Gli3 (Goat polyclonal) | R and D Systems | Cat# AF3690 RRID:AB_2232499 | WB (1:500) |
| Antibody | Anti-β-Actin (clone AC-15) (Mouse monoclonal) | Abcam | Cat# ab6276 RRID:AB_2223210 | WB (1:15,000) |
| Antibody | IRDye 680RD Donkey anti-Goat IgG | LI-COR Biosciences | Cat# 926–68074 RRID:AB_10956736 | WB (1:15,000) |

*Continued*

| Reagent type (species) or resource | Designation | Source or reference | Identifiers | Additional information |
|---|---|---|---|---|
| Antibody | IRDye 800CW Donkey anti-Mouse IgG | LI-COR Biosciences | Cat# 925–32212 RRID:AB_2716622 | WB (1:15,000) |
| Commercial assay or kit | VECTASTAIN Elite ABC-Peroxidase Kit | Vector Laboratories | Cat# PK-6100 RRID:AB_2336819 | |
| Chemical compound, drug | IdU 5-Iodo-2′-deoxyuridine | Sigma Aldrich | Cat# I7125 | (10 mg/ml) |
| Chemical compound, drug | BrdU 5-Bromo-2′-deoxyuridine | Sigma Aldrich | Cat# B5002 | (10 mg/ml) |
| Software, algorithm | Fiji | PMID:22743772? | PRID:SCR_002285 | http://imagej.net/Fiji |
| Software, algorithm | Image Studio Lite | Li-Cor | 4.0 | |
| Software, algorithm | GraphPad Prism | GraphPad Software | 8.4.2 (679) | |
| Software, algorithm | Adobe Photoshop | Adobe Inc | 12.1 | |
| Other | DAPI (4′,6-Diamidino-2-Phenylindole, Dihydrochloride) | Thermo Fisher Scientific | Cat# D1306 RRID:AB_2629482 | IF (1:2000) |

## Mice

All experimental work was carried out in accordance with the UK Animals (Scientific Procedures) Act 1986 and UK Home Office guidelines. All protocols were reviewed and approved by the named veterinary surgeons of the College of Medicine and Veterinary Medicine, the University of Edinburgh, prior to the commencement of experimental work. $Inpp5e^{\Delta}$ ($Inpp5e^{delta}$), $Gli3^{\Delta699}$ ($Gli3^{delta699}$) and $Tctn2^{\Delta}$ ($Tctn2^{tm1.1Reit}$) mouse lines have been described previously (**Bose, 2002**; **Garcia-Gonzalo et al., 2011**; **Jacoby et al., 2009**). $Inpp5e^{\Delta/+}$ mice were interbred to generate $Inpp5e^{\Delta/\Delta}$ embryos; exencephalic $Inpp5e^{\Delta/\Delta}$ embryos which made up ca. 25% of homozygous mutant embryos were excluded from the analyses. Wild-type and $Inpp5e^{\Delta/+}$ litter mate embryos served as controls. $Inpp5e^{\Delta/\Delta};Gli3^{\Delta699/+}$ and $Inpp5e^{\Delta/\Delta};Gli3^{\Delta699/\Delta699}$ embryos were obtained from inter-crosses of $Inpp5e^{\Delta/+};Gli3^{\Delta699/+}$ mice using wild-type, $Inpp5e^{\Delta/+}$ and $Gli3^{\Delta699/+}$ embryos as controls. Embryonic (E) day 0.5 was assumed to start at midday of the day of vaginal plug discovery. Transgenic animals and embryos from both sexes were genotyped as described (**Bose, 2002**; **Jacoby et al., 2009**). For each marker and each stage, three to eight embryos were analyzed.

For measuring cell cycle lengths, pregnant females were intraperitoneally injected with a single dose of IdU (Sigma-Aldrich) (10mg/ml) at E12.5, followed by an injection of BrdU (Sigma-Aldrich) (10 mg/ml) 90 min later. Embryos were harvested 30 min after the second injection. For cell cycle exit analyses, BrdU was injected peritoneally into E11.5 pregnant females and embryos were harvested 24 hr later.

## Immunohistochemistry and in situ hybridization

For immunohistochemistry, embryos were fixed overnight in 4% paraformaldehyde, incubated in 30% sucrose at +4°C for 24 hr, embedded in 30% sucrose/OCT mixture (1:1) and frozen on dry ice. Immunofluorescence staining was performed on 12 to 14 µm cryostat sections as described previously (**Theil, 2005**) with antibodies against Arl13b (mouse) (Neuromab 75–287; 1:1500), rat anti-BrdU (1:50, Abcam #ab6326), mouse anti-BrdU/IdU (B44) (1:50, BD Biosciences #347580), rabbit anti-Cleaved Caspase 3 (1:100, Cell Signaling Technology, #9664), rat anti-Ctip2 (1:1000, Abcam #ab18465), rabbit anti-GFAP (1:1000, Agilent/Dako #Z 0334), rat anti-L1, clone 324 (1:1000, Millipore #MAB5272), rabbit anti-Pax6 (1:400, Biolegend #901301), mouse anti-PCNA (1:500, Abcam

#ab29), rabbit anti-Prox1 (1:1000, RELIA*Tech* #102-PA32). rabbit anti-pHH3 (1:100, Millipore #06–570), mouse anti-Satb2 (1:200, Abcam #ab51502), rabbit anti-Tbr1 (1:400, Abcam #ab31940), rabbit anti-Tbr2 (1:1000, Abcam #ab23345) and rabbit anti-γTUB (Sigma-Aldrich SAB4503045; 1:100). Primary antibodies for immunohistochemistry were detected with Alexa- or Cy2/3-conjugated fluorescent secondary antibodies. The Cleaved Caspase three and Tbr1 signals were amplified using biotinylated secondary IgG antibody (swine anti-rabbit IgG) (1:400, Dako) followed by Alexa Fluor 488 (1:100, Invitrogen) or 568 Streptavidin (1:100, Thermo Fisher Scientific). For counter staining DAPI (1:2000, Thermo Fisher Scientific) was used. Prox1 protein was detected non-fluorescently using biotinylated goat anti-rabbit IgG (1: 400,Agilent (Dako)) followed by avidin-HRP and DAB detection using Vectastain Elite ABC peroxidase kit (Vector laboratories) as described previously (*Magnani et al., 2010*).

In situ hybridization on 12 μm serial paraffin sections were performed as described previously (*Theil, 2005*) using antisense RNA probes for *Axin2* (*Lustig et al., 2002*), *Bmp4* (*Jones et al., 1991*), *Dbx1* (*Yun et al., 2001*), *Dlx2* (*Bulfone et al., 1993*), *Emx1* (*Simeone et al., 1992*), *Gli3* (*Hui et al., 1994*), *Lhx2* (*Liem et al., 1997*), *Msx1* (*Hill et al., 1989*), *Ngn2* (*Gradwohl et al., 1996*), *Nrp2* (*Galceran et al., 2000*), *Pax6* (*Walther and Gruss, 1991*), *Scip1* (*Frantz et al., 1994*), *Wnt2b* (*Grove et al., 1998*).

## Western blot
Protein was extracted from the dorsal telencephalon of E12.5 wild-type and *Inpp5e*$^{\Delta/\Delta}$ embryos (n = 4 samples per genotype) as described previously (*Magnani et al., 2010*). 10 μg protein lysates were subjected to gel electrophoresis on a 3–8% NuPAGE Tris-Acetate gel (Life Technologies), and protein was transferred to a Immobilon-FL membrane (Millipore), which was incubated with goat anti-h/m Gli3 (1:500, R and D Systems #AF3690) and mouse anti-β-Actin antibody (1:15000, Abcam #ab6276). After incubating with donkey anti-goat IgG IRDye680RD (1:15000, LI-COR Biosciences) and donkey anti-mouse IgG IRDye800CW secondary antibodies (1:15000, Life Technologies), signal was detected using LI-COR's Odyssey Infrared Imaging System with Odyssey Software. Values for protein signal intensity were obtained using Image Studio Lite Version4.0. Gli3 repressor and full-length protein levels and the Gli3 repressor/full length were compared between wild-type and mutant tissue using an unpaired t-test.

## Scanning and transmission electron microscopy
TEM and SEM image acquisition were performed in the Cochin Imaging Facility and on the IBPS EM Facility, respectively. For scanning electron microscopy, embryos were dissected in 1.22x PBS (pH 7.4) and fixed overnight with 2% glutaraldehyde in 0.61x PBS (pH 7.4) at 4°C. Heads were then sectioned to separate the dorsal and ventral parts of the telencephalon, exposing their ventricular surfaces. Head samples were washed several times in 1.22x PBS and postfixed for 15 min in 1.22x PBS containing 1% OsO4. Fixed samples were washed several times in ultrapure water, dehydrated with a graded series of ethanol and prepared for scanning electron microscopy using the critical point procedure (CPD7501, Polaron). Their surfaces were coated with a 20 nm gold layer using a gold spattering device (Scancoat Six, Edwards). Samples were observed under a Cambridge S260 scanning electron microscope at 10 keV.

For transmission electron microscopy tissues were fixed for 1 hr with 3% glutaraldehyde, postfixed in 1.22x PBS containing 1% OsO4, then dehydrated with a graded ethanol series. After 10 min in a 1:2 mixture of propane:epoxy resin, tissues were embedded in gelatin capsules with freshly prepared epoxy resin and polymerized at 60°C for 24 hr. Sections (80 nm) obtained using an ultramicrotome (Reichert Ultracut S) were stained with uranyl acetate and Reynold's lead citrate and observed with a Philips CM10 transmission electron microscope.

## Statistical analyses
Data were analyzed using GraphPadPrism eight software with n = 3–8 embryos for all analyses. Shapiro-Wilk normality tests informed whether to use t-tests for normally distributed data and Mann Whitney tests for data which did not pass the normality test. Cortical thickness was analyzed using a two-way ANOVA followed by Sidak's multiple comparisons test. A fisher's exact test was used to analyze the quantification of normal and abnormal cilia. The Gli3 rescue experiments were evaluated

with one way ANOVAS followed by Tukey's multiple comparisons test. A single asterisk indicates significance of $p < 0.05$, two asterisks indicate significance of $p < 0.01$ and three asterisks of $p < 0.001$. Due to morphological changes blinding was not possible and scores were validated by a second independent observer. *Supplementary file 1*-table 1 provides a summary of test statistics.

## Acknowledgements

We are grateful to Drs Thomas Becker, Christos Gkogkas, John Mason, Pleasantine Mill, and David Price for critical comments on the manuscript, and Stéphane Schurmans for the *Inpp5e*$^{\Delta/+}$ mouse line. We also thank Dr Michaël Trichet (electron microscopy platform of the IBPS-Sorbonne Universités Paris 6) and Dr Alain Schmitt (electron microscopy platform of the Institut Cochin CNRS-UMR 8104) for their help with scanning and transmission electron microscopy analyses, respectively. This work was supported by a grants from the Biotechnology and Biological Sciences Research Council (BB/P00122X/1) and from the Simons Initiative for the Developing Brain (SFARI −529085) to TT and from NIH R01GM095941 to JFR.

## Additional information

### Funding

| Funder | Grant reference number | Author |
| --- | --- | --- |
| Biotechnology and Biological Sciences Research Council | BB/P00122X/1 | Thomas Theil |
| NIH Clinical Center | R01GM095941 | Jeremy F Reiter |
| The Simons Initiative for the Developing Brain | SFARI 529085 | Thomas Theil |

The funders had no role in study design, data collection and interpretation, or the decision to submit the work for publication.

### Author contributions

Kerstin Hasenpusch-Theil, Data curation, Formal analysis, Supervision, Validation, Methodology, Writing - review and editing; Christine Laclef, Data curation, Formal analysis, Methodology, Writing - review and editing; Matt Colligan, Formal analysis, Investigation, Writing - review and editing; Eamon Fitzgerald, Formal analysis, Investigation, Methodology; Katherine Howe, Emily Carroll, Shaun R Abrams, Investigation, Methodology; Jeremy F Reiter, Sylvie Schneider-Maunoury, Conceptualization, Supervision, Funding acquisition, Writing - review and editing; Thomas Theil, Conceptualization, Formal analysis, Supervision, Funding acquisition, Validation, Investigation, Writing - original draft, Project administration, Writing - review and editing

### Author ORCIDs

Matt Colligan http://orcid.org/0000-0002-6553-8915
Thomas Theil https://orcid.org/0000-0001-6590-8309

### Ethics

Animal experimentation: All experimental work was carried out in accordance with the UK Animals (Scientific Procedures) Act 1986 and UK Home Office guidelines under the project license numer P53864D41. All protocols were reviewed and approved by the named veterinary surgeons of the College of Medicine and Veterinary Medicine, the University of Edinburgh, prior to the commencement of experimental work.

### Decision letter and Author response

Decision letter https://doi.org/10.7554/eLife.58162.sa1
Author response https://doi.org/10.7554/eLife.58162.sa2

# Additional files

## Supplementary files

- Supplementary file 1. Summary of statistical tests.
- Transparent reporting form

## Data availability

All data generated or analysed during this study are included in the manuscript and supporting files.

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
