## [Decision Letter]

**Acceptance summary:**

This study focuses on the role of Inpp5e, a protein essential for the integrity of primary cilia, in the development of the cerebral cortex. The authors find that mice mutant for Inpp5e have severe morphological defects in the primary cilium of cortical apical Radial Glia Cells, and this is linked to changes in the mode of neurogenesis, favoring direct neurogenesis against indirect neurogenesis via Intermediate Progenitor Cells (IPCs). This effect is nicely shown by studying the abundance of cells positive for marker proteins of IPCs and newborn neurons, as well as cell cycle exit. Intriguingly, this defect is transient during early stages of neurogenesis (circa E12.5), not observable at later stages, but has lasting effects on the abundance of neurons in deep cortical layers. Part of the same defects are found in mouse embryos mutant for Tctn2, another ciliary protein. Then the authors go on to investigate the signaling cascades affected in this mutant and leading to the changes in mode of neurogenesis, and find no changes in Erk but a significant increase in mTOR signaling. Finally, the authors focus on Shh signaling, a well known pathway tightly linked to the primary cilium. They find that the neurogenesis phenotype relates to impaired Gli3 processing and a loss of its repressive form Gli3R. Finally, the authors analyze Gli3D699 mutant mice, where Gli3R is produced independently from the primary cilium, in an Inpp5e mutant background, to show a dose-dependent rescue of the cortical phenotypes, and thus demonstrating that Inpp5e regulates the modes of cortical neurogenesis by regulating Gli3R levels.

**Decision letter after peer review:**

Thank you for sending your article entitled "A transient role of primary cilia in controlling direct versus indirect neurogenesis in the developing cerebral cortex" for peer review at *eLife*. Your article is being evaluated by three peer reviewers, one of whom is a member of our Board of Reviewing Editors, and the evaluation is being overseen by a Reviewing Editor and Marianne Bronner as the Senior Editor.

Given the list of essential revisions, including new experiments, the editors and reviewers invite you to respond as soon as you can with an action plan for the completion of the additional work. We expect a revision plan that under normal circumstances can be accomplished within two months, although we understand that in reality revisions will take longer at the moment. We plan to share your responses with the reviewers and then advise further with a formal decision.

Summary:

This study focuses on the role of Inpp5e, a protein essential for the integrity of primary cilia, in the development of the cerebral cortex. The authors find that mice mutant for Inpp5e have severe morphological defects in the primary cilium of cortical apical Radial Glia Cells, and this is linked to changes in the mode of neurogenesis, favoring direct neurogenesis against indirect neurogenesis via Intermediate Progenitor Cells (IPCs). This effect is nicely shown by studying the abundance of cells positive for marker proteins of IPCs and newborn neurons, as well as cell cycle exit. Intriguingly, this defect is transient during early stages of neurogenesis (circa E12.5), not observable at later stages, but has lasting effects on the abundance of neurons in deep cortical layers. Part of the same defects are found in mouse embryos mutant for Tctn2, another ciliary protein. Then the authors go on to investigate the signaling cascades affected in this mutant and leading to the changes in mode of neurogenesis, and find no changes in Erk but a significant increase in mTOR signaling. Finally, the authors focus on Shh signaling, a well known pathway tightly linked to the primary cilium. They find that the neurogenesis phenotype relates to impaired Gli3 processing and a loss of its repressive form Gli3R. Finally, the authors analyze Gli3D699 mutant mice, where Gli3R is produced independently from the primary cilium, in an Inpp5e mutant background, to show a dose-dependent rescue of the cortical phenotypes, and thus demonstrating that Inpp5e regulates the modes of cortical neurogenesis by regulating Gli3R levels.

The study is nicely planned and performed. However, there are points that must be addressed before the manuscript is ready for publication. In particular:

1) Important clarifications need to be made with regards to the neurogenesis data. Images and illustrations should be reflective of the phenotypes described in the text, and a better discussion is required to reconcile conflicting data.

2) The mTOR/ERK part lacks key controls and is not central to the message of the paper. The section should be removed.

Essential revisions:

1) Signs of cortical distortions, suggestive of patterning defects are seen at the rostral part of the cortex (Figure 1B, F, G). Pax6 expression pattern (gradient) seems also affected in the mutant (decreased Pax6 medially). Please show a more representative image or explain/comment in the legend.

2) Subsection “*Inpp5e* controls direct vs indirect neurogenesis in the lateral neocortex”: The authors state: "these findings show that initially Tbr1+ and later Ctip2+Tbr1- neurons were increasingly formed in the lateral neocortex of Inpp5e embryos". Is that specific to the mutant? It is also the case for control embryos but maybe at a different time point. Please rephrase the sentence.

3) The authors claim that there is an increased neuron production. This does not seem to be the case from Figure 2I, J.

4) Difficult to reconcile results of Figure 2I, J and Figure 3. Please comment and discuss.

5) How can authors conclude that there is no change in the size of the primary cilium from IHC Figure 6 A, B and subsection “Ciliary defects in the forebrain of E12.5 *Inpp5e*^Δ/Δ^ embryos”? This is contradictory to SEM results Figure 6 C-E. Is IHC the best way to assess the size of cilia?

6) The patterning defect (see Supplementary Figure 10 C, D, G H) is such that it makes the results of neurogenesis difficult to interpret.

7) The abundance of IPCs is analyzed by stains against Tbr2/PCNA, and PH3 mitoses at basal position. A decrease in both is interpreted as a loss of IPCs, but in fact the authors find conflicting results in their mutant: loss of Tbr2+PCNA+ cells in the lateral cortex but not medial (Figure 1), and then loss of basal PH3 in medial cortex but not lateral (Figure 3—figure supplement 2).

8) Quantification of PCNA+ cells shows changes in the proportion that are Tbr2+. Importantly, 80% of PCNA+ were Pax6+, and 40% were Tbr2+, so a minimum of 20% of PCNA+ cells are co-expressing Pax6 and Tbr2. Contrary to these results, existing evidence in the field shows that the overlap in Pax6 and Tbr2 expression is really residual in mouse. Please comment on that.

9) Which neuron subpopulations are identified as Ctip2+/Tbr1+ and Ctip2+/Tbr1-?

10) Why increased direct neurogenesis diminishes the overall production of Tbr1 neurons (though higher at E12.5), but with no changes in intermediate and upper layer neurons? This is opposite from what was shown by Cardenas and colleagues in 2018, where forced direct neurogenesis in the early NCx increases deep layer neurons and diminishes upper layer neurons.

11) Subsection “Cortical malformations in *Inpp5e*^Δ / Δ^ embryos” – "This analysis showed that the mutant cortex was thinner laterally but not medially with a more pronounced reduction of the thickness at caudal levels (Figure 4—figure supplement 2)." The data shown in Figure 4—figure supplement 2 contradict this description. In addition, DAPI images are very dark with any detail very difficult to see. These must be better visible, for example by showing in black and white.

12) For the entire Figure 7, pictures and quantification of all variables must include the results from simple lnpp5e mutants. Although these results are shown in earlier figures, they are key to assess the rescue effects of Gli3D699 in hetero and homozygosity, and so they must be presented again in this figure. One clear example of the importance of showing these results here is the phenotype shown (but not mentioned) in Figure 7I, where the dorsal cortex seems to be much shorter than in the control embryo (see the lateral end of the Tbr2+ region, inset box), plus there seem to be three basal ganglia. This is not mentioned as part of the phenotype in lnpp5e mutants. It this caused by this compound genotype?

13) Similar to the previous point, the analysis of Gli3 rescue must include littermates mutant only for, to show if the heterozygous expression of Gli3D699 rescues (or worsens) the deficit in Tbr1+Ctip2+, Tbr1-Ctip2+ and Tbr2+ cells in lnpp5e mutants.

14) Figure 7, panels Q and Q' are at different magnification than the rest of the figure. Pictures of Tbr1 and Ctip2 stains are at insufficient magnification to illustrate the dramatic differences described in the manuscript and quantified in panels P and S.

15) There is a part of the study focused on potential effects in Erk and mTOR signaling. While potentially interesting, this part is completely disconnected from the rest of the study (in fact all the data is in Supplementary Figures), and the authors decide to ignore in the rest of the manuscript, instead focus on Shh signaling. Thus, Erk and mTOR signaling analyses should be removed.

16) Given that other primary cilia mutants do not exhibit deficits in direct versus indirect neurogenesis, as discussed by the authors, they must change the title of their manuscript to indicate that it is Inpp5e that controls this process. Indeed, as also discussed by the authors, this protein plays multiple other roles in brain development (eye, hippocampus, corpus callosum), and thus it is likely to play other roles in cortical development that may contribute to regulate the mode and rate of neurogenesis, independent from the primary cilium.

17) Figure 1—figure supplement 1B,E – PSPB is not visible in KOs, and impossible to assess the existence of a phenotype in this area. These images must be re-framed to make this visible.

18) In Figure 1—figure supplement 3, the authors show some folds in VZ of NCx and hippocampal anlage. Whereas this is not the core of the main findings, the authors must disclose the penetrance and severity of this phenotype.

19) Throughout the manuscript, the authors need to provide the actual numbers for the means in each of their graphs. This could be in the text or figure legend, but in the current version, this data is not present.

20) Could the authors please provide higher magnification images for Figure 1A-J, Figure 2A-F?

21) Figure 3: could the authors indicate that this data is from lateral cortex in the actual figure?

22) In several figures, the cortex is clearly thinner (as the authors indicate in Figure 4—figure supplement 2). However, this seems to be much thinner than would be accounted for by a relatively small decrease in basal progenitor production. Apical progenitor production (S4) and progenitor cell cycle (S5) appears unaffected. Did the authors examine whether there was increased cell death or either progenitors or postmitotic neurons in the mutants?

23) In Figure 4P-U, the cortical layers from the controls and mutants are not aligned. Is this because the overall cortex is thinner and the images are aligned from the basal surface? Including DAPI for these sections would help here.

24) Is the data in Figure 5 from medial or lateral cortex?

25) Why did the authors switch to a parametric t-test for the western blots in Figure 7, S10, S11? Shouldn't a paired non-parametric test (ie: paired samples Wilcoxon) be used here? Also, the data should not be displayed as paired with the lines joining the control and KO samples.

26) For the experiments in S11, the authors show that pAkt/pS6 are elevated Inpp5e-/- mutants, consistent with increased mTOR activity. They attempt to rule this pathway out by showing that treatment with rapamycin for 24hours at E11.5 does not affect the basal progenitor phenotype in Inpp5e-/- mutants. However, the authors do not provide any data showing the effectiveness of their rapamycin treatment (ie, normalization of pAkt/pS6 to wildtype levels). I think they need to be very careful about their interpretation of these results without additional controls.

27) Do the authors have an explanation as to why the phenotypes are predominantly in lateral cortex at E12.5 and medial cortex at E14.5? This seems like something that should be mentioned in the discussion.

28) A model figure at the end would be very useful for explaining how loss of Inppe5 leads to reduced Gli3R levels and alterations in basal progenitor production.

---

## [Author Response]

Essential revisions:1) Signs of cortical distortions, suggestive of patterning defects are seen at the rostral part of the cortex (Figure 1B, F, G). Pax6 expression pattern (gradient) seems also affected in the mutant (decreased Pax6 medially). Please show a more representative image or explain/comment in the legend.

In the original manuscript, we very carefully described patterning defects before we start analysing cortical stem cell development. In brief, the hippocampal anlage is the most affected structure, probably due to reduced Wnt gene expression in the cortical hem, but this structure is not the focus of this study. The neocortex itself is undulated at E12.5 but has a smooth surface by E14.5 and shows a mild patterning defect at the pallial/subpallial boundary. It expresses the progenitor markers Emx1, Lhx2, Pax6 and Ngn2 with reduced Pax6 expression levels in the medial neocortex. We think that these mild patterning defects do not invalidate our analysis and that Figure 1B, F and J are representative of the *Inpp5e* mutant telencephalon.

2) Subsection “Inpp5e controls direct vs indirect neurogenesis in the lateral neocortex”: The authors state: "these findings show that initially Tbr1+ and later Ctip2+Tbr1- neurons were increasingly formed in the lateral neocortex of Inpp5e embryos". Is that specific to the mutant? It is also the case for control embryos but maybe at a different time point. Please rephrase the sentence.

We rephrased the sentence making it clearer that in the lateral neocortex of *Inpp5e*^Δ / Δ^ embryos we observed an increase in the formation of Tbr1+ neurons at E12.5 followed by an augmented generation of Ctip2+Tbr1- neurons at E14.5 (subsection “*Inpp5e* controls direct vs indirect neurogenesis in the lateral neocortex”).

3) The authors claim that there is an increased neuron production. This does not seem to be the case from Figure 2I, J.

In our analysis of neuron formation at E14.5 (Figure 2I-N) we distinguish between two neuronal subpopulations, a Tbr1+/Ctip2+ and a Tbr1-/Ctip2+ population. We show that there is only an increase in the proportion of Tbr1-/ Ctip2+ population in the lateral neocortex which can be clearly seen in the high magnification insets (Figure 2I’, I’’, J’ and J’’). This might not be so obvious from the overview picture (Figure 2I and J) but one needs to consider that the mutant cortex is thinner and we counted cell proportions rather than absolute neuron numbers. To make our point clearer, we used arrows and arrowheads to indicate the different neuron populations in Figure 2I’, I’’, J’ and J’’. We also re-worded the sentence to make it clearer that proportions of neurons are affected (subsection “*Inpp5e* controls direct vs indirect neurogenesis in the lateral neocortex”).

4) Difficult to reconcile results of Figure 2I, J and Figure 3. Please comment and discuss.

We think there is a misunderstanding. Figure 2I, J show the proportion of Tbr1+ and Ctip2+ neurons in E14.5 embryos while Figure 3 shows a cell cycle exit experiment in E12.5 embryos. Due to the different ages and types of experiments, these two figures are not directly comparable.

5) How can authors conclude that there is no change in the size of the primary cilium from IHC Figure 6 A, B and subsection “Ciliary defects in the forebrain of E12.5 Inpp5e^Δ/Δ^ embryos”? This is contradictory to SEM results Figure 6 C-E. Is IHC the best way to assess the size of cilia?

We agree with the reviewer that immunofluorescence does not allow us to measure the size of cilia and we have removed this formulation from the revised manuscript (subsection “Ciliary defects in the forebrain of E12.5 *Inpp5e*^Δ / Δ^ embryos”).

6) The patterning defect (see Supplementary Figure 10 C, D, G H) is such that it makes the results of neurogenesis difficult to interpret.

This comment is very similar to point #1. As outlined above, we have carefully analysed patterning in the *Inpp5e* mutant and think that the mild patterning defects do not interfere with our conclusion on neurogenesis defects in the neocortex. To this specific figure: in the dorsal telencephalon, the most obvious morphological difference in Figure 10 C, D, G, H is a malformation of the hippocampus which is not the focus of this paper. At the rostrocaudal levels we have been investigating the mutant, the neocortex is undulated at E12.5 but smoothens at E14.5 and is otherwise not severely affected. We think that this undulation does not prevent us from investigating the process of direct neurogenesis and the underlying mechanisms. Finally, this figure represents our analysis of Erk signalling but we have removed this analysis from the manuscript as suggested by other reviewers (see point 15).

7) The abundance of IPCs is analyzed by stains against Tbr2/PCNA, and PH3 mitoses at basal position. A decrease in both is interpreted as a loss of IPCs, but in fact the authors find conflicting results in their mutant: loss of Tbr2+PCNA+ cells in the lateral cortex but not medial (Figure 1), and then loss of basal PH3 in medial cortex but not lateral (Figure 4—figure supplement 2).

Tbr2 and pHH3 antibodies label different groups of cells. Tbr2 labels all basal progenitors whereas pHH3 staining identifies mitotic basal progenitors. This different specificity makes it difficult to directly compare the two results but we interpret this finding that the E12.5 basal progenitors in the medial neocortex may have reduced proliferation rates. Indeed, the *Inpp5e* mutants show a reduction in the proportion of basal progenitors in the E14.5 medial neocortex and this reduction may at least partially be explained by the earlier lower proliferation rate. We included this consideration in subsection “*Inpp5e* controls direct vs indirect neurogenesis in the lateral neocortex” of the revised manuscript.

Finally, the reviewer alludes to different effects of the *Inpp5e* mutation on medial and lateral neocortex. We think there are several possible explanations. The phenotypes may follow the neurogenesis gradient with lateral parts of the neocortex being more advanced in basal progenitor and neuron formation. There is also a reduced Pax6 expression in the medial neocortex that is likely to affect neurogenesis. We have included this consideration in the Discussion section.

8) Quantification of PCNA+ cells shows changes in the proportion that are Tbr2+. Importantly, 80% of PCNA+ were Pax6+, and 40% were Tbr2+, so a minimum of 20% of PCNA+ cells are co-expressing Pax6 and Tbr2. Contrary to these results, existing evidence in the field shows that the overlap in Pax6 and Tbr2 expression is really residual in mouse. Please comment on that.

Englund et al., reported that 11.5 +/- 1% of cells co-express Pax6 and Tbr2 in the E14.5 neocortex (Englund et al., 2005), i.e. not too far off the number the reviewer estimated from our Pax6/PCNA and Tbr2/PCNA staining. However, this estimate is derived indirectly from two separate stains and hence might not easily be comparable to a Pax6/Tbr2 double stain.

9) Which neuron subpopulations are identified as Ctip2+/Tbr1+ and Ctip2+/Tbr1-?

We observed that in the E12.5 cortical plate almost all neurons co-express Ctip2 and Tbr1. This population is still present at E14.5 but there is an additional group of neurons which express Ctip2 only, i.e. are Tbr1-/ Ctip2+. Based on the time of their appearance the Tbr1+/Ctip2+ and Tbr1-/Ctip2+ neurons are likely to correspond to layer VI and layer V neurons, respectively, but we do not have definite proof for this. As identifying the fate of these neurons is not the main purpose of this manuscript, we prefer not to go deeper into this issue.

10) Why increased direct neurogenesis diminishes the overall production of Tbr1 neurons (though higher at E12.5), but with no changes in intermediate and upper layer neurons? This is opposite from what was shown by Cardenas and colleagues in 2018, where forced direct neurogenesis in the early NCx increases deep layer neurons and diminishes upper layer neurons.

The reviewer is correct that there is no change in upper layer neuron formation despite the early increase in direct neurogenesis. In the original manuscript, we have already provided a couple of possible explanations. First, the number of basal progenitors has normalized by E14.5 when the majority of upper layer neurons are born. Secondly, there are feedback signals from newly born neurons to radial glial cells to control the sequential production of deep and upper layer neurons. In addition to Notch signalling as described in Cardenas, these signals include *Fgf9* and Neurotrophin 3. Inpp5e could influence this signalling by controlling cilia stability and/or levels of PIP3 which acts as a second messenger in receptor tyrosine kinase signalling. We think these considerations address the reviewer’s concern.

11) Subsection “Cortical malformations in Inpp5e^Δ / Δ^ embryos” – "This analysis showed that the mutant cortex was thinner laterally but not medially with a more pronounced reduction of the thickness at caudal levels (Figure 4—figure supplement 2)." The data shown in Figure 4—figure supplement 2 contradict this description. In addition, DAPI images are very dark with any detail very difficult to see. These must be better visible, for example by showing in black and white.

We are grateful for the reviewer to raise this point. We have amended the text to say that “most of the mutant cortex was thinner except for the rostrolateral level” (subsection “Cortical malformations in *Inpp5e*^Δ / Δ^ embryos”). We have changed the colour of the DAPI staining to black and white (Figure 4—figure supplement 2).

12) For the entire Figure 7, pictures and quantification of all variables must include the results from simple lnpp5e mutants. Although these results are shown in earlier figures, they are key to assess the rescue effects of Gli3D699 in hetero and homozygosity, and so they must be presented again in this figure. One clear example of the importance of showing these results here is the phenotype shown (but not mentioned) in Figure 7I, where the dorsal cortex seems to be much shorter than in the control embryo (see the lateral end of the Tbr2+ region, inset box), plus there seem to be three basal ganglia. This is not mentioned as part of the phenotype in lnpp5e mutants. It this caused by this compound genotype?

In the revised manuscript, we included the analysis of *Inpp5e*^Δ / Δ^;*Gli3*^+/+^ littermates which we obtained from the Gli3 rescue crosses (Figure 7 and Figure 8). In these embryos, we observed a reduced proportion of basal progenitors and an increased fraction of cortical neurons as in *Inpp5e*^Δ / Δ^ embryos derived from the mating of *Inpp5e* heterozygous animals. Moreover, our analysis revealed that the addition of a single Gli3^Δ699^ allele slightly improved the formation of basal progenitors in the E14.5 medial neocortex compared to *Inpp5e*^∆/∆^ embryos but the proportion of basal progenitors was still significantly smaller than in control embryos. These findings support our conclusion that restoring the Gli3 repressor ratio rescues cortical malformations in *Inpp5e*^Δ / Δ^ embryos.

Although the neocortex of *Inpp5e*^Δ / Δ^ ; Gli3 ^Δ699/+^ embryos appears shorter, we can clearly identify the neocortex based on the Tbr1 and Tbr2 staining. At this stage, the three bulges in the ventral telencephalon are specific to and appeared in all E12.5 Inpp5e ^Δ/Δ^;Gli3^Δ699/+^ embryos we analysed; we did not note those in *Inpp5e*^Δ / Δ^ embryos. We will added this information to the legend for Figure 7.

13) Similar to the previous point, the analysis of Gli3 rescue must include littermates mutant only for, to show if the heterozygous expression of Gli3D699 rescues (or worsens) the deficit in Tbr1+Ctip2+, Tbr1-Ctip2+ and Tbr2+ cells in lnpp5e mutants.

As outlined for the previous point, we analysed *Inpp5e*^Δ / Δ^;*Gli3*^+/+^ littermate embryos. The addition of a single Gli3^Δ699^ allele rescued the formation of basal progenitors and Tbr1+ neurons in the lateral neocortex at E12.5 (Figure 7) and led to a slight improvement in the proportions basal progenitors in the E14.5 medial neocortex (Figure 8). The full restoration of basal progenitors and of Tbr1-Ctip2+ neurons in the E14.5 medial neocortex required two copies of the Gli3^Δ699^ allele (Figure 8).

We will perform all analyses and quantifications (Tbr1/Ctip2 and Tbr2/PCNA) on E12.5 and E14.5 Inpp5e^Δ / Δ^;Gli3^+/+^ littermates from the Gli3 rescue experiment as in Figure 7 of the original manuscript.

14) Figure 7, panels Q and Q' are at different magnification than the rest of the figure. Pictures of Tbr1 and Ctip2 stains are at insufficient magnification to illustrate the dramatic differences described in the manuscript and quantified in panels P and S.

We have increased the magnifications of the insets of what is Figure 8 A’, C’, E’ and G’ in the revised manuscript. This and addition of arrows pointing at Ctip2+/Tbr1- neurons should better illustrate the differences in this population. We also ensured that the panels from the different genotypes have the same magnification.

15) There is a part of the study focused on potential effects in Erk and mTOR signaling. While potentially interesting, this part is completely disconnected from the rest of the study (in fact all the data is in Supplementary Figures), and the authors decide to ignore in the rest of the manuscript, instead focus on Shh signaling. Thus, Erk and mTOR signaling analyses should be removed.

We have removed the Erk and mTOR analyses from the manuscript. We agree with the reviewers that the major focus of the manuscript is on Gli3 signalling and the manuscript makes better reading without the Erk and mTOR analyses which appear disconnected.

16) Given that other primary cilia mutants do not exhibit deficits in direct versus indirect neurogenesis, as discussed by the authors, they must change the title of their manuscript to indicate that it is Inpp5e that controls this process. Indeed, as also discussed by the authors, this protein plays multiple other roles in brain development (eye, hippocampus, corpus callosum), and thus it is likely to play other roles in cortical development that may contribute to regulate the mode and rate of neurogenesis, independent from the primary cilium.

We have changed the title and made a corresponding change at the end of the Introduction. Mutations in ciliary genes can have a variety of effects on cortical development (see Discussion section), however, we not only found an increase in direct neurogenesis in the *Inpp5e* mutant but also in the *Tctn2* mutant. This suggests that *Inpp5e*‘s effect on direct neurogenesis is more general and might for example be obscured by more severe patterning defects in other ciliary mutants. Moreover, the other phenotypes mentioned by the reviewer (eye, hippocampus, corpus callosum) are also rescued by re-introducing Gli3R strongly suggesting that these phenotypes are also cilia dependent.

17) Figure 1—figure supplement 1B,E – PSPB is not visible in KOs, and impossible to assess the existence of a phenotype in this area. These images must be re-framed to make this visible.

We think there must be a misunderstanding. Figure 1—figure supplement 1B, E focusses on the formation of the corticoseptal boundary not the PSPB as indicated in the figure legend. PSPB formation is analysed in Figure 1—figure supplement 1G-L with higher magnification insets clearly showing scattered Pax6+ and *Dlx2* expressing cells.

18) In Figure 1—figure supplement 3, the authors show some folds in VZ of NCx and hippocampal anlage. Whereas this is not the core of the main findings, the authors must disclose the penetrance and severity of this phenotype.

We observed these folds which occur with 100% penetrance only at caudal positions, i.e. at the rostral/caudal level of the thalamus. Moreover, they become more prominent at more caudal levels. We have added this information in the revised manuscript (subsection “*Inpp5e*^Δ / Δ^ embryos show mild telencephalic patterning defects”) but we would like to emphasize that this interesting observation is not the main focus of the manuscript. Indeed, these folds complicate the analysis of neurogenesis defects in the caudal neocortex. For this reason, we have not included this region in our analysis.

19) Throughout the manuscript, the authors need to provide the actual numbers for the means in each of their graphs. This could be in the text or figure legend, but in the current version, this data is not present.

In the originally submitted manuscript, we included an Excel file (Supplementary file 1) providing a summary of descriptive statistics of all tests used in this manuscript. We prefer this approach as it increases the readability of the main text but also provides the interested reader with much more information on statistical test results than we could include in the main text. We should have referred to this table in the original manuscript but have done this in the revised version at the end of the statistics paragraph (Materials and methods section). We will, however, follow the reviewer’s advice to include mean numbers in the main text if they prefer us to do so.

20) Could the authors please provide higher magnification images for Figure 1A-J, Figure 2A-F?

In the revised manuscript we included representative higher magnification images for Figure 1 and Figure 2.

21) Figure 3: could the authors indicate that this data is from lateral cortex in the actual figure?

We can confirm that this data is from lateral cortex and we have amended the figure legend correspondingly.

22) In several figures, the cortex is clearly thinner (as the authors indicate in Figure 4—figure supplement 2). However, this seems to be much thinner than would be accounted for by a relatively small decrease in basal progenitor production. Apical progenitor production (S4) and progenitor cell cycle (S5) appears unaffected. Did the authors examine whether there was increased cell death or either progenitors or postmitotic neurons in the mutants?

We have examined cell death but found very few apoptotic cells in the cortex of both, control and mutant embryos. This new data is included in Figure 3—figure supplement 1.

23) In Figure 4P-U, the cortical layers from the controls and mutants are not aligned. Is this because the overall cortex is thinner and the images are aligned from the basal surface? Including DAPI for these sections would help here.

The reviewer is correct, the overall cortex is thinner at caudal levels (for quantification see Figure 4—figure supplement 2) and images in Figure 4P-U are aligned from the basal surface. We included this information already in the figure legend of the original manuscript. Unfortunately, the high power pictures were taken without the DAPI channel.

24) Is the data in Figure 5 from medial or lateral cortex?

We can confirm that this data is from the lateral cortex, this information was added to the figure legend.

25) Why did the authors switch to a parametric t-test for the western blots in Figure 7, S10, S11? Shouldn't a paired non-parametric test (ie: paired samples Wilcoxon) be used here? Also, the data should not be displayed as paired with the lines joining the control and KO samples.

We realized that the background normalisation of the Western blot using an automatic function of the LI-COR software was not adequate. The protein lysates are derived from single dorsal telencephali and therefore have a low protein concentration. As a consequence, we needed to load large volumes of the lysates which might have caused some smiling, especially for the Gli3R bands. To take this effect into account, we repeated the background normalisation by placing a user defined box into each lane. The original blot and how we evaluated the blot is shown in Author response image 1:

Testing the new data set for normality (Shapiro-Wilk test) and for equal variance revealed that the data is normally distributed and that there is no significant difference in variance between the control and mutant data sets. For this reason, we used an unpaired t-test to statistically evaluate this analysis.To use the above approach consistently throughout the paper, we tested all our data sets for normal distribution and equal variance. To evaluate statistical significance, we used unpaired t-tests for normally distributed data showing equal variance and non-parametric tests otherwise. In this way, we employed statistical tests consistently throughout the paper. This procedure is added to the Material and methods section. We would like to emphasize that this did not change the outcomes except for a significantly reduced proportion of mitotic RGCs in the E12.5 medial neocortex (Figure 3—figure supplement 2A, C). which does not, however, affect our conclusions. Moreover, we have summarized all details of the statistical tests in Supplementary file 1.

26) For the experiments in S11, the authors show that pAkt/pS6 are elevated Inpp5e-/- mutants, consistent with increased mTOR activity. They attempt to rule this pathway out by showing that treatment with rapamycin for 24 hours at E11.5 does not affect the basal progenitor phenotype in Inpp5e-/- mutants. However, the authors do not provide any data showing the effectiveness of their rapamycin treatment (ie, normalization of pAkt/pS6 to wildtype levels). I think they need to be very careful about their interpretation of these results without additional controls.

As suggested in the editor’s comments, we have removed the analysis of Akt and mTOR signalling in the revised manuscript.

27) Do the authors have an explanation as to why the phenotypes are predominantly in lateral cortex at E12.5 and medial cortex at E14.5? This seems like something that should be mentioned in the discussion.

A similar issue was already raised under point #7 and, as outlined in more detail in our response to this point, this differential effect may be due to the lateral to medial gradient of neurogenesis and/or gene expression changes ( for example Pax6). We have included this consideration in the Discussion section.

28) A model figure at the end would be very useful for explaining how loss of Inppe5 leads to reduced Gli3R levels and alterations in basal progenitor production.

We have included a model (Figure 9) summarizing our findings on the different mutants used in this manuscript. This new figure supports the discussion in subsection “*Inpp5e* controls direct/indirect neurogenesis through Gli3 processing” of how loss of *Inpp5e* functions leads to reduced Gli3 R levels and changes in direct and indirect neurogenesis.